# An outside individual option increases optimism and facilitates collaboration when groups form flexibly

Ryutaro Mori [1,2], Nobuyuki Hanaki [3,4] & Tatsuya Kameda [5,6,7,8] ✉

Voluntary participation is a central yet understudied aspect of collaboration. Here, we model collaboration as people's voluntary choices between joining an uncertain public goods provisioning in groups and pursuing a less profitable but certain individual option. First, we find that voluntariness in collaboration increases the likelihood of group success via two pathways, both contributing to form more optimistic groups: pessimistic defectors are filtered out from groups, and some individuals update their beliefs to become cooperative. Second, we reconcile these findings with existing literature that highlights the detrimental effects of an individual option. We argue that the impact of an outside individual option on collaboration depends on the "externality" of loners − the influence that those leaving the group still exert on group endeavors. Theoretically and experimentally, we show that if collaboration allows for flexible group formation, the negative externality of loners remains limited, and the presence of an individual option robustly aids collaborative success.

Ranging from business ventures to academic research, collaboration is a robust strategy in human societies to achieve objectives that can never be accomplished by any individual working alone[1–3]. However, initiating successful collaborations is not at all trivial, as it requires costly and coordinated efforts from multiple individuals[3–6]. Specifically, each individual has an incentive to exploit the collective output without incurring personal effort, discouraging many from committing to collaboration. Presumably because the major difficulty stems from this free rider problem, collaboration has been mainly modeled as public goods provisioning. There, individuals are bound within a group with fixed memberships and must decide whether to cooperate with the group. Several mechanisms have been identified to mitigate the free rider problem, including other-regarding preferences[7–9], internalized norms[10,11], peer punishment[12–15], and reciprocity in repeated interactions[16–21].

Yet, such models have largely sidelined the fact that many collaborations in the real world do not involve the entire public or take place within predetermined group boundaries (but see literature of optional public goods game[22–25] for important exceptions). In reality, people often have individual alternatives outside groups and are thus free to opt in or out of group endeavors. In the case of a startup company, for instance, only those who voluntarily choose to join will work together. Consequently, even though the company still faces the issue of free riding, those who opt not to participate in the collaborative venture remain uninvolved.

To understand group collaboration in such voluntary situations, we extend previous models of public goods provision and coordination problems using threshold public goods game[26–35] (Fig. 1A lower left). In a group, each member chooses either to contribute their endowment to the group or to keep it for themself. Only if a sufficient

[1]Department of Social Psychology, The University of Tokyo, 7-3-1 Hongo, Bunkyo-ku, Tokyo 113-0033, Japan. [2]Japan Society for the Promotion of Science, 5-3-1 Kojimachi, Chiyoda-ku, Tokyo 102-0083, Japan. [3]Institute of Social and Economic Research, Osaka University, 6-1 Mihogaoka, Ibaraki-shi, Osaka 567-0047, Japan. [4]University of Limassol, 21 Glafkou Kleride Avenue 2107, Aglandjia, Nicosia, Cyprus. [5]Faculty of Mathematical Informatics, Meiji Gakuin University, 1518 Kamikurata-cho, Totsuka-ku, Yokohama-shi, Kanagawa 244-853, Japan. [6]Center for Interdisciplinary Informatics, Meiji Gakuin University, 1-2-37 Shirokanedai, Minato-ku, Tokyo 108-8636, Japan. [7]Center for Experimental Research in Social Sciences, Hokkaido University, N10W7, Kita-ku, Sapporo, Hokkaido 060-0810, Japan. [8]Brain Science Institute, Tamagawa University, 6-1-1 Tamagawagakuen, Machida-shi, Tokyo 194-8610, Japan. ✉e-mail: tkameda@mi.meijigakuin.ac.jp

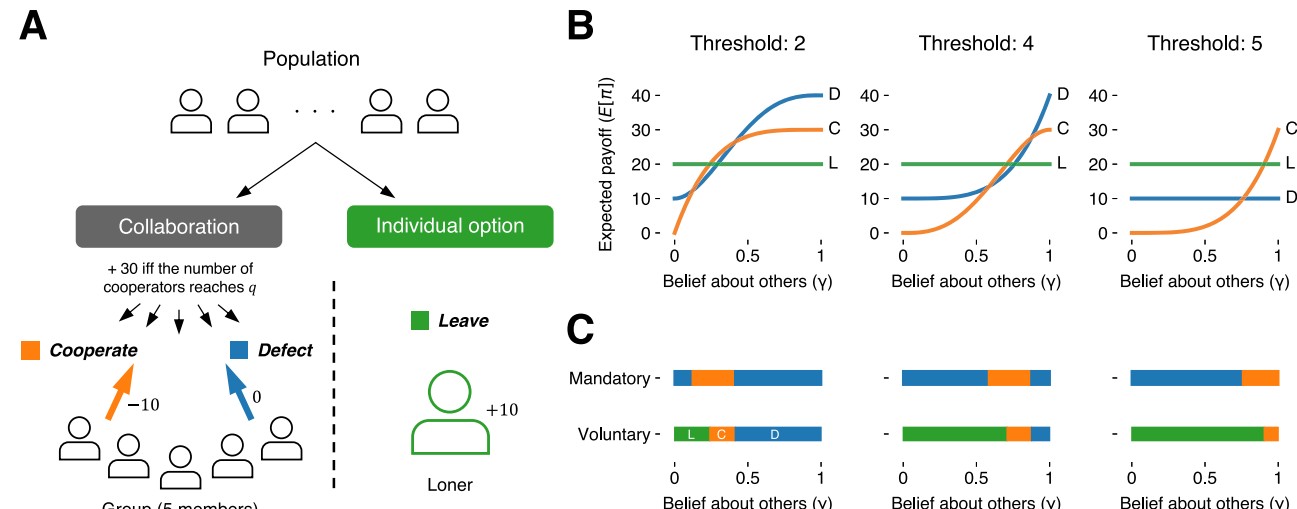

**Fig. 1 | Modeling collaboration under voluntary participation. A** Individuals can either opt in to group collaboration or choose an individual option outside of the group. When opting to collaborate, players are randomly assigned to groups of five members (if the total number of players opting in is not a multiple of five, any remaining players are assigned to the individual option). Group members can choose whether to cooperate (incurring the private cost of 10 points) on the group project. If the number of cooperators is equal to or more than the predetermined threshold value, $q$ ($q = 2$, 4, or 5), then all the group members earn 30 extra points. When choosing the individual option, players are guaranteed to earn a smaller additional payoff of 10 points irrespective of other players' choices. Expected payoff (**B**) and best response (**C**) as a function of subjective belief about how likely others are to cooperate within groups (i.e., $\gamma$). Each color represents a possible action under mandatory or voluntary participation: orange = cooperate (C), blue = defect (D), and green = leave (L). Recall that under mandatory participation, where leaving is not available, the best response is derived by comparing between the expected payoffs of cooperation and defection.

number (i.e., threshold) of individuals have cooperated will the group succeed in producing the public goods that are to be shared equally among its members. Crucially, we contrast this uncertain but potentially rewarding group collaboration with a certain but less profitable individual option outside the group: Players are free to pursue an individual option outside of groups or to participate in group collaboration with others who have also voluntarily opted in (Fig. 1A).

Recently, an influential line of research[31–33,36] employed a similar (but distinct, as will be discussed later) setting and experimentally demonstrated that the presence of individual options can have detrimental effects on group collaboration. The researchers argued that just as billionaires may not be highly motivated to contribute their share to the social insurance system, the option of self-reliance crowds out collaborative motivations, and groups may have to struggle even more to get enough contributions.

Here we challenge this view, arguing that the availability of an individual option renders collaboration voluntary and thereby facilitates, rather than hinders, successful group collaboration. We first develop benchmark predictions about the difference in collaborative success under the presence or absence of the individual option (i.e., voluntary vs. mandatory participation in the group). Specifically, we analyze a simple model where players form subjective beliefs about other players' actions (i.e., whether to cooperate) and choose actions conditioned on the belief. To test the predictions, we then proceed to a preregistered behavioral experiment. In accordance with the modeling results, we found that compared to mandatory participation, voluntary participation increased the rate of individual cooperation within groups, up to the point where most groups succeeded in producing collective benefits. Further, two distinct behavioral mechanisms—pessimistic defectors being filtered out from groups (i.e., self-selection of optimists into groups)[37] and some pessimists switching from defection to cooperation via improved subjective beliefs about others' cooperativeness—together operate under voluntary participation, contributing to form groups with a higher level of optimism and increasing the frequency of cooperators within groups.

Finally, we argue that an apparent divergence between the positive (current study) and the negative[31–33] effects of individual alternatives to collaboration can be synthesized by explicitly considering the effect of the loners' "externality" on the collective outcome: the varying degrees of negative impact that players who opt for an outside individual option still exert on group endeavors. We show that when the collaboration concerns public goods for a fixed group, the large externality of loners can worsen collaboration, whereas when the collaboration concerns public goods with flexible members under voluntary participation, an individual option indeed facilitates collaborative success.

## Results

### Modeling collaboration using a voluntary threshold public goods game

We consider the threshold public goods game with an individual outside option for both theoretical development and experiments (Fig. 1A). All players start with an endowment of 10 points. They first decide whether to participate in a group-based collaboration or to employ an individual option outside of the group ("leave"). Choosing the individual option secures 10 extra points irrespective of other players' decisions, plus the initial endowment (10 points). When choosing to participate in collaboration, players are randomly assigned to groups of five members and play the threshold public goods game (in the experiment, if the total number of participants opting for groups is not a multiple of five, the remaining participants are assigned to the individual option). In groups, each member chooses whether to invest their initial endowment of 10 points in the group ("cooperate") or to keep it for themselves ("defect"). If the number of cooperators reaches a predetermined threshold, all members of the group (including defectors) earn 30 extra points. If the group fails to gather enough cooperators, the collaboration does not create any gains, and the investments are not returned. Players are commonly informed of these rules at the outset, including the threshold value for group collaboration.

Critically, the presence/absence of the individual option corresponds to voluntary/mandatory participation in collaboration, respectively. The threshold value, $q$, determines the minimal number of cooperators for successful collaboration and thereby alters the

degree of free-riding incentive within groups. With a higher threshold, collaboration requires more cooperators and involves less temptation to free-ride, as we see below. Here, we consider three threshold values: $q = 2$, 4, or 5 (out of 5), representing a strong, weak, or null temptation to free-ride, respectively. By manipulating these two factors orthogonally with a 2 (voluntary vs. mandatory participation) × 3 (threshold values) factorial design, we study how voluntary participation affects collaborative efforts under varying degrees of free-riding incentives within the collaboration.

## Model analysis: whether/how voluntary participation can aid collaboration

In the threshold public goods game, the action that maximizes the player's own payoff depends on other players' actions. Thus, we start by assuming that a player forms a subjective belief about others' average cooperativeness (i.e., how likely it is that other group members will cooperate in a given situation) and selects their own action according to that belief [34,35]. We denote the player's belief about others' cooperativeness as $\gamma \in [0,1]$, their action as $x \in \chi$, and the resultant payoff as $\pi$. When collaboration is mandatory, the set of possible actions for the player, $\chi$, is {C,D} and when collaboration is voluntary, it is {C, D, L} (where C = cooperate, D = defect, and L = leave).

Given their belief about others, the expected payoff for each action ($E[\pi|x = \cdot]$) becomes

$$
\begin{aligned}
E[\pi|x = \text{C}] &= 30 \times \Gamma_{q-1}, \\
E[\pi|x = \text{D}] &= 10 + 30 \times \Gamma_q, \\
E[\pi|x = \text{L}] &= 10 + 10
\end{aligned}
\tag{1}
$$

where $\Gamma_k$ denotes the probability that at least $k$ of four other members will cooperate, according to the binomial expansion with their belief $\gamma$: $\Gamma_k = \sum_{j=k}^{4} \binom{4}{j} \gamma^j (1-\gamma)^{4-j}$. Figure 1B displays expected payoffs as a function of $\gamma$ under three threshold values ($q = 2$, 4, or 5 from the left). Figure 1C further depicts the best responses (i.e., actions maximizing expected payoff given belief) under mandatory and voluntary participation.

Two observations are noteworthy. First, when participation is mandatory, defection arises as the best response in two separate regions (see blue intervals in Fig. 1C: Mandatory, left and middle). These regions may be interpreted as the operation of two different psychological motives [29,38]. One is "fear" that cooperation may fall short even if the player themself cooperates, and it operates when the player expects little cooperation from others (i.e., small $\gamma$). The other can be termed "greed": When a player expects a great deal of cooperation from others (i.e., large $\gamma$), they may greedily defect to free ride on successful collaboration. Note that with the threshold of 5 (Fig. 1C, right), greedy defection cannot exist, as free riding is no longer possible (i.e., a player's own defection guarantees failed group endeavor), effectively transforming the game into a pure coordination game.

Second, when participation is voluntary, the configuration of the best response changes drastically. As seen in the green intervals in Fig. 1C: Voluntary, leaving is the best possible action across the thresholds unless the player's $\gamma$ is sufficiently high. In other words, those who hold low expectations about their peers' cooperation will naturally choose the individual option outside the group. These loners will cover "pessimistic defectors" who would not have cooperated within groups out of fear under mandatory participation and may also include some potential cooperators who hold relatively low expectations. As such, voluntary participation will work as a self-selection mechanism that filters pessimistic individuals out of collaborative efforts.

To reasonably predict the cooperation rate within groups (among those who opt for collaboration) resulting from the individual best response function (Fig. 1C), we further need to consider the distribution of subjective beliefs in the population, $\phi(\gamma)$. The proportion of the population that chooses action $x(= \{C, D, L\})$, $r_x$, can be obtained by integrating the best response with respect to $\gamma$:

$$
r_x = \int_0^1 I\{\text{Best response is } x, \text{given } \gamma\} \phi(\gamma) d\gamma,
\tag{2}
$$

where $I\{A\}$ is the indicator function (returns 1 if $A$ is true and 0 otherwise). Accordingly, the cooperation rate in the population, $p_{\text{coop}}$, is obtained as follows:

$$
p_{\text{coop}} = \frac{r_C}{r_C + r_D}.
\tag{3}
$$

Let us now compare the cooperation rate between voluntary and mandatory participation. Consider first when the threshold value is 5. Here, if participation is voluntary (see Fig. 1C: Voluntary, right), there is no interval of $\gamma$ where the best response is defection, which leads to $r_D = 0$ irrespective of the distribution of beliefs, $\phi(\gamma)$. Therefore, the resultant cooperation rate, $p_{\text{coop}}$, equals 1 under voluntary participation (as long as $r_C > 0$). In contrast, with smaller threshold values (see Fig. 1C: Voluntary, left and middle), where there remain possibilities for greedy defection even under voluntary participation, the results should depend on $\phi(\gamma)$. We investigated how $p_{\text{coop}}$ differs between mandatory and voluntary participation under various specifications of $\phi(\gamma)$. Without losing much generality, we can model $\phi(\gamma)$ with a Beta distribution and calculate $p_{\text{coop}}$ while changing its parameters systematically. We found that the resultant $p_{\text{coop}}$ is higher under voluntary participation than under mandatory participation across a wide range of parameters of $\phi(\gamma)$ (see Supplementary Fig. 1). Overall, we can predict that voluntary participation will induce the self-selection of optimists into groups and of pessimists into individual options, which consequently increases the population-level cooperation rate for group endeavors.

We have considered how voluntary participation may alter the member composition of group endeavors via self-selection dynamics. Notice that, thus far, we have assumed that the distribution of players' beliefs is fixed, whether participation is mandatory or voluntary. In other words, we have argued that players may choose different actions in these two situations (e.g., defecting/cooperating vs. leaving according to the best response function in Fig. 1C) but keep their subjective beliefs ($\gamma$) intact. However, some players may engage in reasonings similar to ours and further update their beliefs about others' cooperativeness, which potentially increases (or decreases) cooperation under voluntary participation. Indeed, when the threshold is 5, our analysis above suggests that players can reasonably predict that no one will defect under voluntary participation, because players are always better off leaving than defecting (Fig. 1B, C: Voluntary, right). Thus, one can argue that in the voluntary condition, players come to expect 100% cooperation from other members who participate in groups and then decide themselves to behave cooperatively.

However, with smaller threshold values (Fig. 1B, C: Voluntary, left and middle), the predictions become less clear because of the possibility of greedy defection. At first, players may take the self-selection dynamics into account, forming more optimistic beliefs. Yet, they may further update their subjective beliefs recursively, fluctuating between optimism and pessimism. For instance, a player who initially anticipates increased cooperation via self-selection may end up with pessimism by reasoning that other players will have the same anticipation and begin free-riding with the updated beliefs.

If we consider only the equilibrium case where players' beliefs and actual plays ultimately converge under best responses, voluntary participation is predicted to induce greater cooperation than

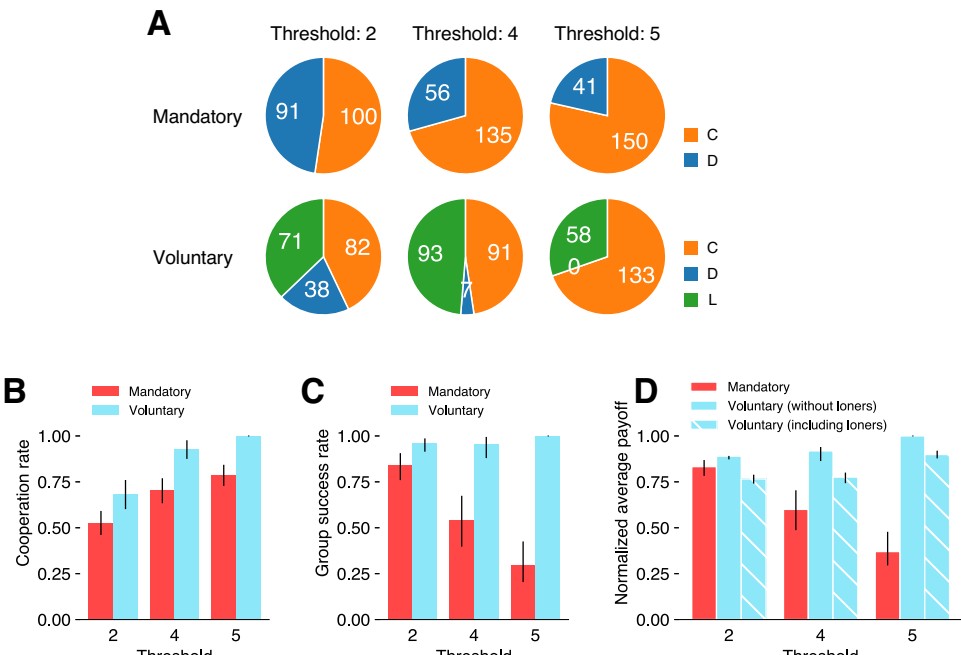

**Fig. 2 | Voluntary participation supports collaborative success. A** Breakdown of participants choosing each action in each condition. The pie charts depict the number of participants who selected each action (orange: cooperate [C], blue: defect [D], and green: leave [L]; $n = 191$ individuals), with exact numbers in the wedges. **B** Individual cooperation rate [#C/(#C + #D); Mandatory: $n = 191$ individuals in every threshold; Voluntary: Threshold 2: $n = 120$; 4: $n = 98$, 5: $n = 133$], (**C**) group success rate (same $n$s as in (**B**)), and (**D**) normalized (i.e., ratio against the highest possible payoff for each threshold) average payoff in each condition (Mandatory or Voluntary (including loners): $n = 191$ individuals in every threshold; Voluntary (without loners): Threshold 2: $n = 120$; 4: $n = 98$; 5: $n = 133$). In each of the three panels, the x-axis indicates threshold values, and the color coding indicates whether participation in groups is mandatory (red) or voluntary (cyan). Error bars in (**B**–**D**) indicate 95% bootstrapped confidence intervals (CIs), with the bar heights showing the exact values from the original experimental data. See Methods for the exact procedures to calculate the CIs. In (**D**), plain cyan bars show the average payoff under voluntary participation just within groups (excluding loners), and hatched cyan bars show the average payoff for the entire population with loners included.

mandatory participation (see Supplementary Note 1). In a nutshell, because a noncooperative equilibrium of groups where no one cooperates yields a lower return than the outside individual option, it is unsustainable and eliminated under voluntary participation. Nevertheless, it is unclear a priori whether, or to what extent, people change their subjective expectations between mandatory and voluntary participation, especially under the smaller threshold values ($q = 2, 4$). We thus leave this as an empirical question to be addressed with our experimental data and revisit this issue in the Discussion.

To sum up, our theoretical analysis suggests two related but distinct mechanisms by which voluntary participation may facilitate collaboration. First, as a baseline, it will make optimists self-select into groups and pessimists into the individual option, increasing the frequency of cooperators within groups. The second possible mechanism is the updating of beliefs: Anticipating self-selection, individuals may further update their subjective beliefs and possibly become more optimistic about successful collaboration. Together, we conjecture that voluntary participation works as a natural device to filter pessimistic individuals (who would otherwise defect from fear) out of groups and/or encourage them to choose cooperation.

### Experimental tests of the predictions
To test the predictions, we ran a preregistered behavioral experiment ($N = 191$). We had around 30 (30–35) participants in each experimental session. In each session, we manipulated (1) whether players had the individual option (i.e., voluntary or mandatory participation) and (2) the threshold value (2, 4, or 5) of the public goods game, resulting in a $2 \times 3$ factorial within-subject design. Participants played each of the six different conditions once in a pseudo-randomized order with no feedback until the end of the whole experiment. Within each condition, participants first estimated how likely others would be to choose

each action and then selected their own actions in the game. Participants' own action selections and estimations about others' actions were both incentivized (see Methods for details of the experimental procedure). Additionally, we elicited participants' economic and psychological characteristics that may partially account for heterogeneities in their play, including risk preference[39] and inequity aversion[8,40], which will be addressed exploratively in Supplementary Note 2.

### Voluntary participation improved success rates and efficiencies of group collaboration
Figure 2A shows the exact number of participants choosing each action in each condition. Responding to the difference in the threshold values and whether participation was mandatory or voluntary, participants indeed changed their action selections. Figure 2B displays participants' cooperation rate for the public good within groups in each condition. Note that here, as defined in Eq. (3), cooperation rate refers specifically to the frequency of cooperators among those who choose to participate in groups [#C/(#C + #D)] rather than the frequency among the entire population [#C/(#C + #D + #L)], as it is the former frequency that determines the outcome of group collaboration. The cooperation rate is higher under the voluntary conditions compared to the mandatory conditions across the three thresholds (the difference in cooperation rate; $\Delta p_{coop}$: $q = 2$: $\Delta p_{coop} = 0.16$, bootstrapped 95% confidence interval [95% CI; hereafter, all CIs refer to bootstrapped 95% CIs; see Methods for the details of the bootstrapping procedure] [0.06, 0.25]; $q = 4$: $\Delta p_{coop} = 0.22$, 95% CI [0.14, 0.31]; $q = 5$: $\Delta p_{coop} = 0.21$, 95% CI [0.16, 0.27]). Voluntary participation elevated the cooperation rates by around 20 points for each threshold.

To evaluate how the increase in individual cooperation rate affected the outcomes of groups, we next looked at population-

level statistics. One important measure is how likely groups are to succeed in generating the collective benefits ("group success rate"; probability of a group gathering enough cooperators; $p_{success} = \sum_{k=q}^{5} (0ex5k) p_{coop}^k (1 - p_{coop})^{5-k}$). As seen in Fig. 2C, under the mandatory conditions, the group success rate decreased as the threshold rose; only 29.9% of groups would succeed with the highest threshold of 5. In contrast, in the voluntary conditions, the group success rate reached nearly 1 across the thresholds. For each threshold, the group success rates were significantly higher in the voluntary conditions compared to the mandatory conditions, with the gaps being particularly prominent under higher threshold values (the difference in group success rate ($\Delta p_{success}$): $q = 2$: $\Delta p_{success} = 0.12$, 95% CI [0.05, 0.20]; $q = 4$: $\Delta p_{success} = 0.41$, 95% CI [0.27, 0.56]; $q = 5$: $\Delta p_{success} = 0.70$, 95% CI [0.57, 0.80]).

Another key population-level metric is the average payoff, or efficiency. A rise in cooperation rates within the voluntary groups does not necessarily result in greater efficiency for two distinct reasons: overcooperation within groups and overpresence of loners outside of groups. Overcooperation could reduce the average payoff because contributions above the threshold do not generate any further gain for the group. Also, if there are too many loners whose payoffs (20 points) are lower than those of successful group members (30 points for cooperators or 40 points for defectors), that could also cause the average payoff to drop at the population-level.

Fig. 2D displays the average payoff normalized as a ratio against the most efficient net payoff possible in a group (i.e., with the exact threshold number of cooperators). Here, we distinguished between two efficiencies: the average payoff within groups while excluding loners and the average payoff for the entire population with loners included. The two differ under voluntary participation (plain or hatched cyan bars), but coincide (red bars) under mandatory participation (as there are no loners by definition). Notice that the potential issue of overcooperation can be assessed by comparing voluntary and mandatory participation in terms of the average payoff within groups, and the issue of too many loners by the average payoff for the entire population. Results indicate that in the former comparison, the average payoff was higher in the voluntary conditions than in the mandatory conditions across the thresholds (the difference in efficiency within groups: $q = 2$: 0.06, 95% CI [0.02, 0.10]; $q = 4$: 0.32, 95% CI [0.20, 0.43]; $q = 5$: 0.63, 95% CI [0.52, 0.71]). In the second comparison, the average payoff for the entire population was also higher in voluntary conditions except for $q = 2$ (the difference in efficiency including loners: $q = 2$: −0.07, 95% CI [−0.11, −0.01]; $q = 4$: 0.18, 95% CI [0.07, 0.28]; $q = 5$: 0.53, 95% CI [0.43, 0.60]). For $q = 2$, more than 80% of the mandatory groups were already successful (see Fig. 2C left), yielding a much higher average payoff to their members (30 or 40 points) than that of loners (20 points).

Taken together, the results indicate that voluntary participation did improve the efficiency among group members while avoiding overcooperation issues and could also improve the efficiency of the entire population despite the issue of too many loners.

## Two mechanisms underlying the collaborative success under voluntary participation

Having established that voluntary participation leads to higher cooperation rates and generally better collective outcomes than mandatory participation, we now turn to its mechanisms. Recall that we conjectured two possible pathways: Pessimistic players who would defect out of fear under mandatory participation will be filtered out from groups (i.e., will opt for the individual option) and/ or form more optimistic beliefs and turn to cooperation within groups.

To address the first point, we examined how participants' defection rates in the mandatory condition and leaving rates in the voluntary condition both correlate with their "original" beliefs about others'

cooperativeness (γ in the mandatory condition). Figure 3A shows that under each threshold, originally more pessimistic participants (i.e., holding a smaller γ) were more likely to defect (blue) when placed in the mandatory condition (mixed-effects logistic regression: $z = -3.92$, $β = -7.74$, $p < 0.001$, 95% CI [−10.26, −6.32]; see Methods for details) and to leave (green) groups for the individual option in the voluntary condition (mixed-effects logistic regression: $z = -4.45$, $β = -1.73$, $p < 0.001$, 95% CI [−2.54, −1.05]). Figure 3B displays proportions of participants who left groups in the voluntary condition, as a function of their action in the mandatory condition with the same threshold values. Defectors in the mandatory condition (darker green) were more likely to leave groups compared to cooperators (lighter green), across the thresholds under voluntary participation (mixed-effects logistic regression: $z = 4.86$, $β = 1.11$, $p < .001$, 95% CI [0.68, 1.59]). Hence, voluntary participation indeed worked as a self-selection mechanism via subjective beliefs, filtering pessimistic individuals (who were disproportionately defectors rather than cooperators) out of collaborative groups.

Next, we consider the second pathway. Can voluntary participation lead some pessimistic defectors to form more optimistic beliefs and turn to cooperation within groups, rather than merely prompting them to opt out from groups? To examine this point, we now focus on participants who stayed in groups under both voluntary and mandatory conditions in each threshold (i.e., non-loners). Removing the loners, the cooperation rates within groups were still significantly higher in the voluntary condition than in the mandatory condition under the threshold values of 4 and 5 (Fig. 3C; the difference in cooperation rates among non-loners: $q = 4$: 0.09, 95% CI [0.01, 0.18]; $q = 5$: 0.14, 95% CI [0.09, 0.20]). When the threshold is 2, the difference was mostly positive but contains zero ($q = 2$: 0.08, 95% CI [−0.02, 0.19]). These results suggest that there was a net cooperative action shift among those who kept opting in to groups; participants were more likely to switch from defection in the mandatory condition to cooperation in the voluntary condition than the opposite.

Then, did this cooperative action shift parallel the optimistic updating of beliefs, as we conjectured? Fig. 3D displays scatter plots of each participant's subjective beliefs about others' cooperativeness, γ, in the mandatory (x-axis) and the voluntary (y-axis) conditions. Here, each dot represents a non-loner who stayed in groups, with its position indicating the belief change and its color indicating the action change (see the caption of Fig. 3D for details). As seen in the figure, for each threshold value, individuals who showed positive (cooperative) action change (orange) were primarily distributed above the diagonal, becoming more optimistic in the voluntary condition. The relationship between the within-individual changes in action and belief was statistically significant (mixed-effects regression: $z = 9.12$, $β = 0.82$, $p < 0.001$, 95% CI [0.65, 1.00]), confirming that the cooperative action changes from the mandatory to voluntary conditions were accompanied by optimistic belief changes at the individual level. Together, these results suggest that, beyond the effect of self-selection (filtering out pessimistic defectors), voluntary participation encourages a significant number of individuals to develop optimism regarding others' cooperativeness and turn to cooperation within groups.

It is also important to notice that across the six conditions, participants' defection rates decreased almost monotonically as their expectations about others' cooperativeness increased (blue lines in Fig. 3A for the mandatory conditions; see Supplementary Fig. 2 for the voluntary conditions). Standard theories of expected utility maximization dictate that players should respond to the probability of their own decisions being pivotal in the threshold public goods game (i.e., both necessary and sufficient for the provision of collective benefits[30]). Yet, in our experimental data, raw expectations (i.e., "how likely are others to cooperate?") were more

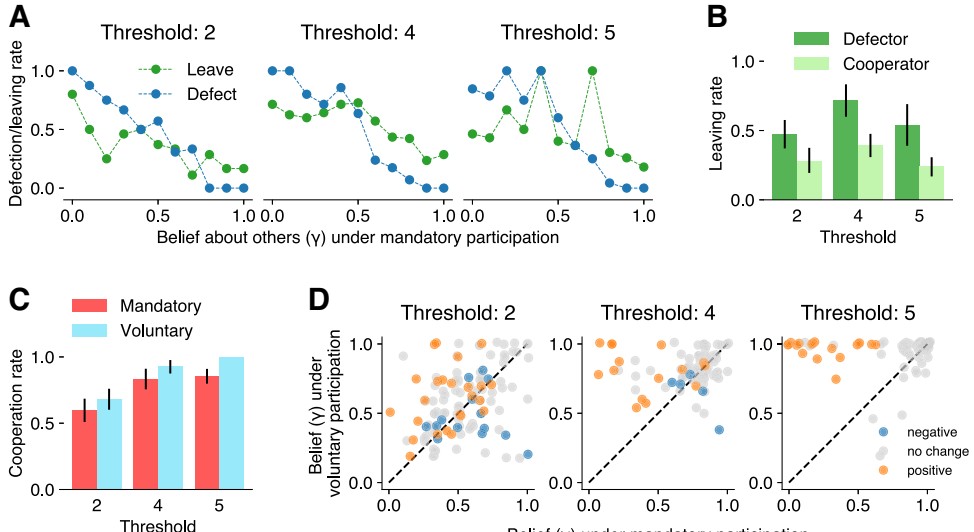

**Fig. 3 | Voluntary participation filters out or encourages pessimistic defectors.**
**A** Rates of participants' defection in the mandatory condition (blue) and leaving in the voluntary condition (green), as a function of their original beliefs about others' cooperativeness (γ in the mandatory conditions; grouped in 0.1 increments).
**B** Proportions of participants who left groups in the voluntary condition as a function of their actions in the mandatory condition with the same threshold values ($n = 191$ individuals for each threshold). Defectors in the mandatory condition (darker green) were more likely to leave groups compared to cooperators (lighter green), across the thresholds under voluntary participation. **C** Non-loners' cooperation rates in the two conditions (red: mandatory, cyan: voluntary; Threshold 2: $n = 120$ individuals, 4: $n = 98$, 5: $n = 133$). For the calculation of cooperation rates in

the mandatory condition, we included only those participants who stayed in groups in the voluntary condition of the same threshold (i.e., non-loners). Error bars in (**B**, **C**) indicate 95% bootstrapped CIs, with the bar heights showing the exact values from the original experimental data. See Methods for the exact procedures to calculate the CIs. **D** Scatter plots of participants' subjective beliefs about others' cooperativeness (γ) in the mandatory (*x*-axis) and voluntary (*y*-axis) conditions for each threshold (Threshold 2: $n = 120$ individuals; 4: $n = 98$; 5: $n = 133$). Dot colors correspond to the changes in action (orange: positive change where a player changed action from defecting in the mandatory condition to cooperating in the voluntary condition, gray: no change, blue: negative change to defection). As in (**C**), we display solely the data from non-loners.

predictive of their actions (whether to cooperate or defect) than the pivotal probabilities calculated from them, across conditions (see Supplementary Note 3 and Supplementary Fig. 3 for the analysis of receiver operating characteristics). These results underscore the importance of fear, not greed, as the primary driver of defection within groups attempting to initiate collaboration under social uncertainty[29,38,41]. Consequently, for successful collaboration, it is more critical to mitigate pessimistic defection than to prevent greedy defection.

## Theoretical mapping: reconciling seemingly contradictory effects of individual options

With the theoretical and experimental analyses, we have demonstrated the *positive* impacts of voluntary participation (i.e., the presence of an outside individual option) on collaborative success. However, as we mentioned earlier, an influential line of research has recently highlighted the *negative* effects of individual options on public goods provisioning in groups[31–33,36]. How can we reconcile the seemingly conflicting results? Here, we propose an integrative view highlighting the varying degrees of externality (impact) that loners, who opt for individual solutions, still have on the outcome of group endeavors.

In ref. 31, Gross and De Dreu confronted participants with a variant of the threshold public goods game called the collective risk social dilemma[42,43]. In their game, participants are embedded in groups of four or five members ("village") and face a shared risk of flooding to their village that causes the loss of all properties. Participants can choose (a) investing their endowment to build a public dam that surrounds the entire village, (b) investing to build a personal dam only around their house, or (c) keeping their endowment to themselves without investing in any dam. By manipulating the cost–benefit ratio associated with constructing the dam collectively (option a) versus individually (option b), they examined how participants shift

from collective to individual options. Crucially, at the intermediate levels of the cost–benefit ratio, where the uncertain collective solution can add only relatively little efficiency over the certain individual option, participants struggled to balance between self-reliance and interdependence: Some started to employ the individual solution while others continued to attempt to solve the problem collectively, leading to a lower probability of collective success as well as a decrease in populational efficiency. Notice that, in their set-up, with more loners who choose the individual solution, a smaller number of villagers must share the cost to build the public dam that surrounds the entire village including the loners' houses; consequently, in terms of provision of the public dam, loners function essentially the same as defectors who retain their endowment and do not fund any dam. That is since group boundaries are fixed from the outset irrespective of the individuals' decision to opt for a collective or individual solution, the cooperation rate determining the collective outcome is the proportion of cooperators among the entire population in the village (fixed group).

In contrast, in our scenario focusing on collaboration[2], the cooperation rate determining the collective outcome for groups is the proportion of cooperators among players who opt in to groups. This is the natural consequence of our assumption that participation in collaboration is voluntary rather than mandatory. Group boundaries are not fixed from the outset but flexible, and group formation comes after the individuals' decision to opt in or not; loners are excluded from the groups at the time of their formation. Note that this does not imply that people no longer suffer from the free-riding problem: Groups still must create shared benefits from the costly efforts of some members.

To facilitate a finer comparison, we introduce a new parameter $\rho \in [0, 1]$ reflecting the degree of loners' externality on the collective outcome, as illustrated in Fig. 4A. The cooperation rate, which effectively determines whether the group can produce collective goods, is

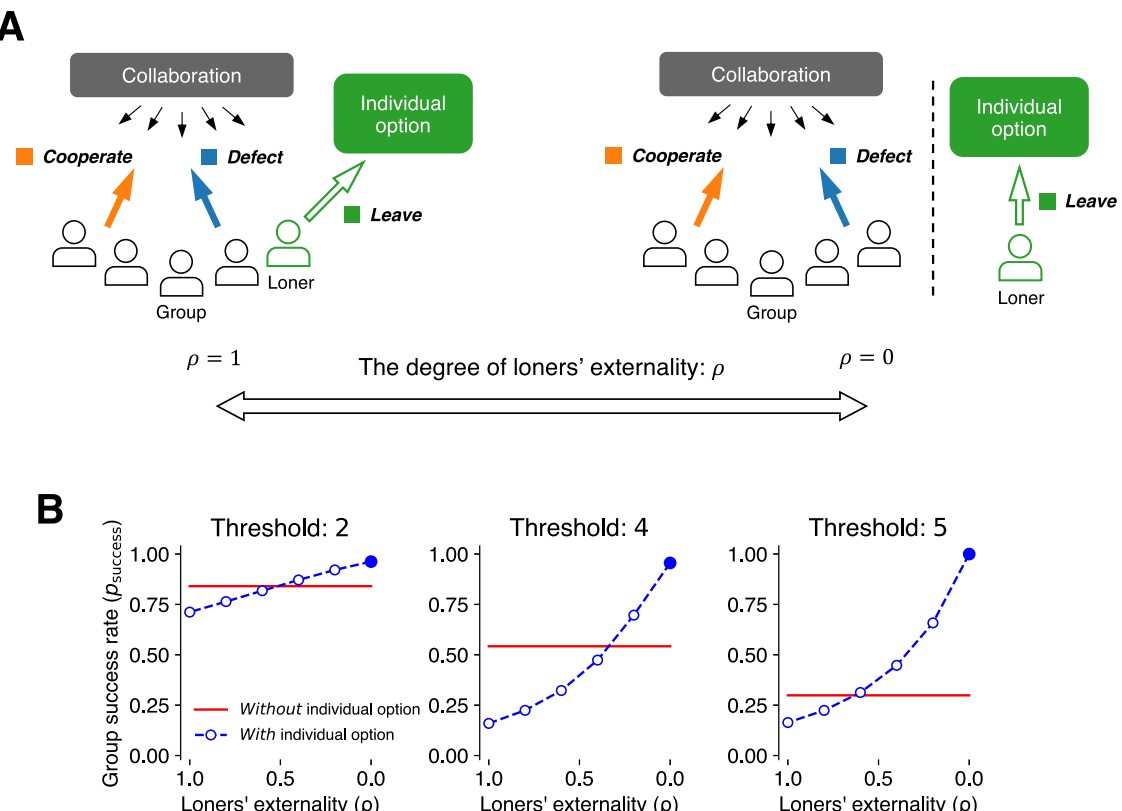

**Fig. 4 | Synthesizing positive and negative effects of the individual option on collaborative success via the degree of loners' externality. A** Left: When groups are fixed in advance (e.g., the entire population is a group), loners have full externality on the collective outcome and effectively function the same as defectors to the group (e.g., the "village" example in ref. 31). Right: When group boundaries are flexible, groups consist only of individuals who voluntarily opt in, and thus loners have no externality on the collective outcome. **B** Group success rate (computed from participants' choice data, $r_C$, $r_D$, and $r_L$, in the main experiment) as a function of the degree of loners' externality, $\rho$. Blue lines correspond to the situation *with* the individual option (voluntary participation) and red to the situation *without* the individual option (mandatory participation). Note that, for the sake of consistency with (**A**), loners' externality on the *x*-axis of graphs is smaller toward the right. C Cooperate, D defect, L leave.

expressed as a function of $\rho$:

$$p_{coop}(\rho) = \frac{r_C}{r_C + r_D + \rho r_L}. \tag{4}$$

Assuming the group size is 5, the group success rate is computed as

$$p_{success}(\rho, q) = \sum_{k=q}^{5} \binom{5}{k} p_{coop}^{k}(\rho)\left(1 - p_{coop}(\rho)\right)^{5-k}, \tag{5}$$

where $q$ is the threshold value.

The two scenarios presented above can be seen as the opposite extremes of the continuum. When loners have full externality on the collective outcome and function identically to defectors ($\rho = 1$; Fig. 4A left), as in the village case[31], the effective cooperation rate determining the collective outcome is $\frac{r_C}{r_C + r_D + r_L} = r_C$. In contrast, when loners are separated from collaboration and thus exert no externality ($\rho = 0$; Fig. 4A right), the collective outcome will be determined by the ratio of cooperators to individuals who opt in to collaboration: $\frac{r_C}{r_C + r_D}$.

Similarly, there should be cases where loners exert a partial externality on the group outcome. For example, when group members express an intention to leave, they may not be immediately replaced by new entrants and may end up remaining in the group temporarily, possibly because of the group's limited ability to recruit others or even

as a part of a formal contract. These cases correspond to intermediate values of $\rho$ ($0 < \rho < 1$).

In Fig. 4B, we computed the group success rate as a function of loners' externality ($\rho$), by aggregating participants' choice data from the main experiment with Eqs. (4) and (5). The figure illustrates several key results. When $\rho = 1$ (left endpoint), the group success rate with the individual option (blue) is lower than that without the individual option (red), corroborating the argument that introducing an individual alternative handicaps public goods provisioning when the group boundary is fixed. However, with a smaller $\rho$ (toward the right endpoint), the relationship reverses. This reaffirms the main claim of this study, namely, that an individual alternative coupled with voluntary participation to groups rather aids collaboration (Fig. 2B).

Note that the individual choice data used to construct Fig. 4B were obtained from the zero-externality scenario ($\rho = 0$) of the voluntary conditions in the main experiment. Given that variations in the loners' externality are likely to affect not only the aggregation method (Eq. (4)) but also the participants' action selections themselves, restriction of the individual choice data just under $\rho = 0$ may affect the prospect of the theoretical analysis.

To explore to what extent the group success rate is negatively affected by larger $\rho$, we conducted an additional experiment. Employing the same protocol as in the voluntary condition of the main experiment with the threshold at 4, here, we manipulated the loners' externality at three levels ($\rho = 0, 0.5, $ or $1$; see Methods for details). The results are shown in Fig. 5. The participants indeed changed their

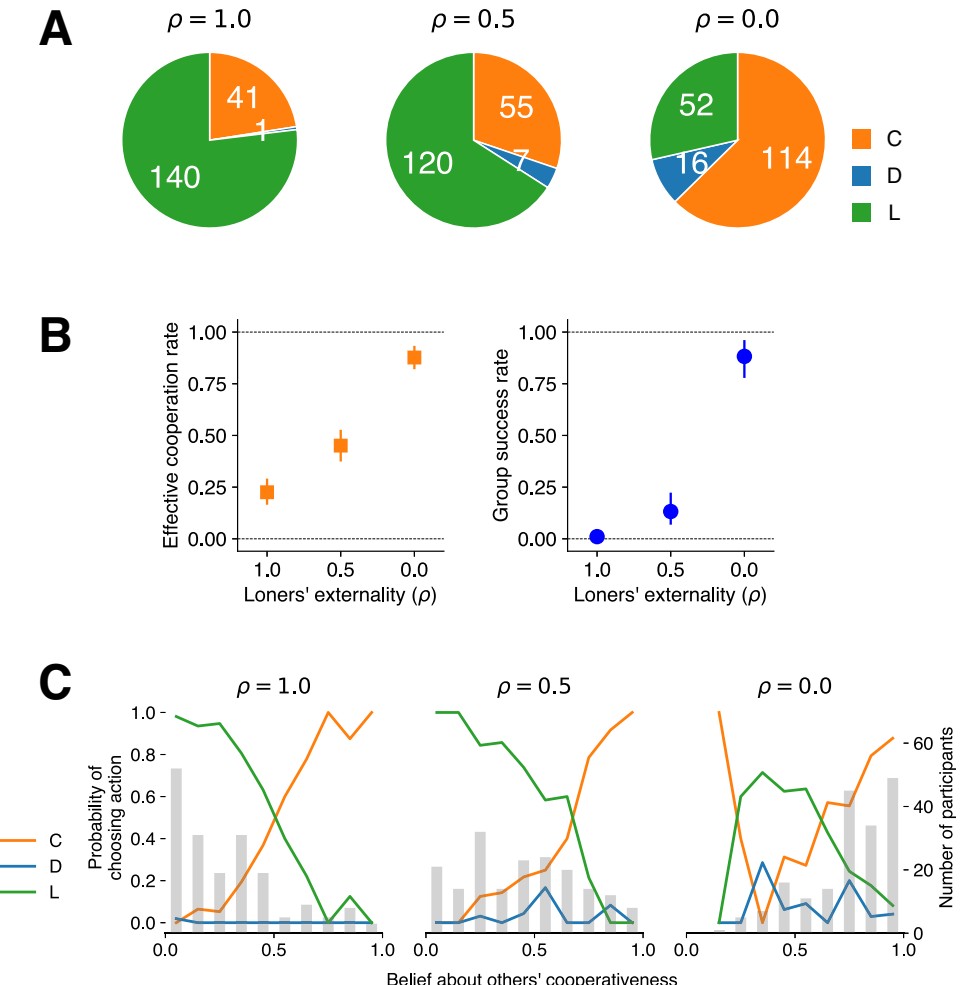

**Fig. 5 | Results from the additional experiment, manipulating loners' externality (ρ) on group outcome. A** Number of participants who chose each action (cooperate [C]: orange, defect [D]: blue, or leave [L]: green) in each ρ condition. Note that ρs are placed inversely from right (0) to left (1) for the sake of consistency with Fig. 4. **B** Effective cooperation rate ($r_C / (r_C + r_D + \rho r_L)$; left) and group success rate (right) as a function of ρ. Error bars indicate bootstrapped 95% CIs ($n = 182$ individuals), with the midpoints showing the exact values computed from the experimental data. See Methods for the exact procedures to calculate the CIs. **C** The distributions of participants' beliefs about (effective) cooperation rate (gray bars) and probability of players choosing each action conditioned on the beliefs (color lines; C: orange, D: blue, or L: green). As participants reported their beliefs about the number of other participants (of 30) choosing each action, beliefs about the effective cooperation rate were computed based on Eq. (4).

action selections (cooperate, defect, or leave), responding to the differences in ρ. As ρ increased, the sheer number of participants choosing to cooperate or to defect decreased, while those opting for leaving increased (Fig. 5A). Notably, the resultant effective cooperation rate (Eq. (4)) and the group success rate (Eq. (5)) were higher at lower ρ values (Fig. 5B); very high group success (88.4%) at ρ = 0, which we observed in the main experiment (95.6%; Fig. 2B: threshold = 4), was replicated. We observed that the drops in cooperation and group success were already large at ρ = 0.5. This drop is attributed not just to the aggregation method (Eq. (4)) but also to a significant increase in the number of loners from 28.6% (at ρ = 0) to 65.9% (at ρ = 0.5; Fig. 5A). These patterns suggest that players' perception of loners' small externality at the time of decision making is an important requirement for an individual option to support collaborative success. Also, of note, these differences in action selections across ρ values were accompanied by differences in beliefs about effective cooperation rates (Eq. (4)); participants were significantly more optimistic about others' cooperativeness at smaller ρ values (Fig. 5C gray histogram; a mixed-effects logistic regression: $z = -16.24$, $\beta = -3.55$, $p < 0.001$, 95% CI $[-3.98, -3.14]$)). Further, the mapping pattern from beliefs to decisions was also replicated, confirming the monotonic increase in

cooperation rate and decrease in leaving rate as a function of beliefs about others' cooperativeness (Fig. 5C orange and green lines).

Summarizing the theoretical and experimental results, the seemingly contradictory claims about the impact of an outside individual option on collaborative success can be integrated through the varying degrees of loners' externality on the collective outcome, ρ.

## Discussion

Numerous studies have examined what behavioral tendencies, cognitive abilities, or bottom-up norms are needed to overcome difficulties associated with collaboration, as well as evolutionary pathways explaining why humans could have equipped themselves with such dispositions[7-21,38]. Yet, focusing attention on the free-riding problem, most research has largely left out the mundane observation that collaborative efforts often take place among flexible group members who have gathered voluntarily, except for several key studies we discuss below.

The current study has addressed this gap by modeling collaboration as a voluntary threshold public goods game with an outside individual option. Using the game, we analyzed how voluntary participation can aid collaboration by promoting optimism among group

participants. As conjectured, evaluating the outcome of the whole population, voluntary participation increased the group success rate (Fig. 2C) as well as efficiency (i.e., average payoff; Fig. 2D). The results also confirm that this increase in collaborative success was underscored by (1) the self-selection of optimistic individuals into groups and (2) the formation of optimistic expectations among some of the otherwise pessimistic defectors (Fig. 3).

Here, we discuss the positioning of the current study in relation to the few key studies that have also explored the consequences of introducing an individual option. First, theoretical and behavioral results consistent with the first self-selection mechanism have been reported in other settings[37,44,45]. For example, Orbell and Dawes[37] had participants play a one-shot prisoner's dilemma game under two conditions: a binary-choice condition, where the two players were obliged to choose between cooperation and defection, and a trinary-choice condition, with an additional exit option (more specifically, when one player selected the exit option, both players received a payoff lower than that of mutual cooperation but higher than that of mutual defection). Observing that the cooperation rate among formed pairs was greater in the trinary condition, they argued that cooperators are less likely to exit than defectors. Although their use of a between-subjects design (rather than a within-subject design, as we used) seems to obscure this interpretation, their logic is similar to ours.

Most interestingly, however, the current study further revealed the second mechanism whereby some pessimistic defectors formed more optimistic beliefs about others' cooperativeness under voluntary participation, opting for cooperation within groups instead of becoming loners (Fig. 3C, D). Several previous studies using the repeated interaction paradigm operationalized the exit option as the possibility to change partners and demonstrated that, if coupled with some reputation mechanism, it can create an additional incentive for cooperation[46–53]. However, in our study, the increase in cooperation was driven by the intrinsic belief changes among participants, as opposed to the extrinsic behavioral-control system (i.e., partner selection under a reputation mechanism) in those schemes.

Another line of research that addresses the potential merits of individual outside options is the optional public goods game, most prominently introduced by Hauert, Brandt, and colleagues[22–25]. They argued that as more individuals exit from the group where a linear public goods game is played, the return to a remaining player's own contribution (marginal per capita return) increases to the point where cooperation is more beneficial than defection—if enough players choose to leave, the game ceases to be a social dilemma anymore. Through evolutionary models, they showed that the population is not dominated by defectors but usually oscillates with cooperators, defectors, and loners coexisting in the long run. Note, however, in our setting with a threshold public goods game, the number of individuals choosing the individual option does not alleviate the social dilemma; groups still face the issue of free riding even though loners are voluntarily excluded from the public goods. Moreover, unlike the evolutionary analysis that concerns populational dynamics over time among agents each following a predetermined fixed strategy[22–25], our model assumes that agents change behaviors flexibly in response to changes in beliefs. We have empirically verified that participants indeed updated their subjective beliefs in response to the absence or presence of an individual option, even in one-shot decision scenarios. We believe that these critical distinctions highlight the complementarity of our investigation to the previous studies.

Finally, in relation to the studies by Gross and De Dreu[31] on the negative effects of loners, we argue that whether outside individual options aid or hinder collaboration depends on the degree of loners' externality on the collective outcome. We have illustrated this point with a simple model (Fig. 4), and the additional experiment showed that as the loners' externality approached zero, the group success rate did increase to close to 1, as suggested by the model (Fig. 5). Certainly,

some of the large-scale societal challenges, such as combating climate change and sustaining the healthcare system in a country, inevitably involve everyone in the society and thus have inflexible boundaries. However, we note that many new collaborative opportunities that we encounter in our daily lives, neither involve the entire population nor occur within predetermined group boundaries; groups form through voluntary participation and consist exclusively of individuals who voluntarily opt in. Consequently, loners have little to no externality on such group endeavors.

There are several limitations about the scope of this paper. First, although this study focused on the idea that self-selection occurs on the basis of expectations about others' cooperativeness, there may be other cognitive or motivational processes that influence self-selection in group endeavors. For example, if competence or confidence varies among individuals, different self-selection dynamics may operate, depending on specific incentive structures[54–58]. It may also be plausible that human individuals are motivated by factors other than payoff maximization[8–10,35], such as efficiency and fairness concerns (but see also Supplementary Note 2 and Supplementary Table 1 for the irrelevance of fairness concerns in our main experiment). We believe that our game incorporates the minimal incentive structures underlying collaborative situations and thus provides benchmark predictions of what can happen under voluntary and mandatory participation, as well as a solid starting point for future investigations of other possible cognitive and motivational aspects.

Second, our model does not explicitly predict how individuals form their subjective beliefs. We first assume that these beliefs remain intact regardless of whether the group participation is voluntary or mandatory (Fig. 1B, C), and then examine equilibrium prediction, where beliefs ultimately converge with the distribution of players' actions (Supplementary Note 1). We took this approach because we found it intractable to explicitly model belief formation given that it depends heavily on additional individual factors. These factors may include players' depth of thought and their perceptions of others' decision-making noise—aspects that are known to vary widely among individuals and contexts[59–61]. Nevertheless, we acknowledge the potential value and interest in fully endogenizing players' beliefs.

Relatedly, our study focuses on one-shot situations where decisions are made only once. The mechanisms that promote cooperation in one-shot cases rely on the variation of subjective expectations among players. This is not necessarily the case in repeated situations, as players observe their action histories in common and may adjust their expectations until they converge. Future research should investigate whether voluntary participation helps people maintain, not just initiate, cooperation in repeated interactions.

Last, our investigation with the threshold public goods game with outside individual option does not necessarily provide a general model, in that it does not comprehensively explore key parameters, including the group size, potential benefits of successful collaboration, and loner benefits. We partially extended our model analysis by relaxing the specific assumptions about these parameters employed in the experiment (see Supplementary Note 4 and Supplementary Figs. 4, 5) and found that it yielded mostly the same results as in our original analysis. Carefully extrapolating the findings for broader parameter regions would be a fruitful future direction.

Humans are frequently confronted with difficulties in and functionalities of collaboration. The current study suggests that the existence of an outside individual option renders participation in collaboration voluntary and can encourage individuals to pursue the collaborative endeavor more optimistically.

## Methods
### Main experiment
We recruited 206 participants from the subject pools of the University of Tokyo (Japan) and Meiji Gakuin University (Tokyo, Japan) in

February 2022. We report results from a total of 191 participants, excluding a participant who afterward declared that she had already graduated from the university and 14 participants who failed to participate in the experiment on time. Of the remaining 191 participants, 86 were male, 100 were female, 1 chose "other," and the remaining 4 participants declined to answer. The mean age of participants was 22.8 years ($SD = 2.7$). For about 60 min of participation, participants were paid 2124 JPY ($SD = 316$) on average ($M = 18.47$ USD, $SD = 2.74$). The experiment was approved by the Ethics Committee of the University of Tokyo, and every participant provided informed consent.

The intended sample size included variables, and main hypotheses of the main experiment were preregistered on the Open Science Framework (https://doi.org/10.17605/OSF.IO/PDNWT) on February 1st, 2022, prior to the collection of any data.

We ran a total of six experimental sessions that lasted about an hour. In each session, around 30 (min: 30, max: 35) participants enrolled in the experiment from their own computers while being connected via Zoom. During the experiment, participants were kept anonymous and were not permitted to communicate with each other. At the beginning of each session, the experimenter read aloud the overall instructions to all participants while they viewed the instructions on their respective screens. We instructed participants that the monetary reward would be the sum of a constant completion fee (1200 JPY) and a bonus based on their performance during the experiment. After giving informed consent, participants played the main task, proceeded to two additional tasks eliciting their risk and social preferences, and answered a postexperimental questionnaire.

For the main task, there were six conditions, using a 2 (group participation: mandatory or voluntary) × 3 (threshold value: 2, 4, or 5) factorial within-subject design. All participants played each of the six conditions once. Participants were explicitly instructed that they were playing the game with around 30 other participants engaging in the experiment simultaneously. In each condition, subjects first read brief instructions about the rules of the respective game and took comprehension quizzes, during which they could ask the experimenter any questions via chat in Zoom. After answering all the comprehension quizzes correctly, they proceeded to the actual play. In the game, participants first estimated other participants' actions ("How many of 30 other participants do you think will choose to cooperate, defect, or leave, respectively?"), indicated their confidence in their estimate on a scale of 0 to 100, and then decided on their own actions. There was no feedback about other participants' decisions or resultant payoffs until the end of the whole experiment. The order of the six conditions was (partially) randomized across participants: Half of the participants played all three voluntary conditions first while the other half played all three mandatory conditions first, with the order of threshold values within the voluntary and mandatory conditions being randomized.

We incentivized estimations about others' actions as well as participants' own action selections. It was emphasized during the instructions that the bonus reward for the main task was set to increase as the participants estimated other participants' actions more accurately and as they acquired more points from their own actions. Specifically, the bonus in the main task, $v_i$, was determined randomly by either the participant's estimation accuracy or the acquired points in one randomly selected condition:

If the estimation accuracy is selected, $v_i = 800 - \frac{80}{6}$

$\times \sum_{x \in X} \left| \frac{e_{i,x}}{30} - \frac{\sum_j I\{x_j = x\}}{\sum_j 1} \right|$, and if the acquired point is selected, $v_i = 20 \times \pi_i$,

where $e_{i,x}$ is the participant's estimation about the number of other participants choosing the focal action $x$, $\sum_j I\{x_j = x\}$ corresponds to the actual number of participants (other than the participant themself) choosing the action, and $\pi_i$ represents the points the participant

earned in the focal condition. Note that for the estimation question, regardless of the exact number of actual participants other than the player themself (which ranged from 29 to 34), we asked participants to estimate actions of 30 others. The accuracy was then determined by comparing the ratio of each action to the total, as shown in the equation. The bonus was set to range from 0 to 800 JPY, regardless of whether the estimation accuracy or the number of acquired points was chosen.

After completing the six conditions in the main task, participants proceeded to two additional incentivized tasks designed to elicit their risk and social preferences, respectively. Specifically, they were asked to choose which lotteries to take and how to share a sum of money with another participant. From the participants' answers to the lottery questions, we estimated their risk preference parameters assuming constant relative risk aversion for their utility functions. From the answers to the sharing questions, we estimated their social preference using Fehr and Schmidt's inequity aversion utility function[32]. Both tasks were incentivized by telling participants that their monetary bonus would be determined by their decision on one randomly selected question for each task (see the Supplementary Information). At the end of the experiment, participants were paid the sum of the participation fee, the bonus from the main task, and the bonus from the additional task. Refer to the github repository for more detailed protocols and the instruction slides (translated into English) used in the main experiment.

## Additional experiment

We recruited 182 participants from the subject pool of the Institute of Social and Economic Research (ISER) at Osaka University in January 2024. Of the 182, 110 were male, 70 were female, and 2 chose not to answer. The mean age of participants was 22.9 years ($SD = 2.8$). For about 30 min of participation, they were paid 974 JPY on average. The constant completion fee was smaller (500 JPY) than in the main experiment, reflecting the shorter duration. We ran a total of six experimental sessions, each consisting of around 30 (min: 28, max: 32) participants enrolling in the experiment simultaneously. The experiment was approved by the Ethics Committee of the ISER (No. 20240102).

The main task of participants was the threshold public goods game with the option to "leave" groups. Employing the same protocol as in the main experiment with the threshold at 4 (for five members), we manipulated the impact (externality) of the decision to leave on group outcome at three levels. Specifically, we set the probability of a loner being involved in the group assignment at one of three levels: $\rho = 0, 0.5, \text{or} 1$. When $\rho = 0$, the formation of five-person groups was restricted to players who did not choose to leave. Conversely, when $\rho = 1$, the assignment to groups included everyone regardless of their choice of whether to leave. At the intermediate level of $\rho = 0.5$, half of the individuals who chose to leave were also included randomly in the five-person group assignments. Those who chose to leave but were nevertheless included in groups earned just the loners' payoff. However, they functioned identically to defectors in terms of the group outcome in that they made zero contribution to group welfare while occupying a seat in the five-member group. Notice that this manipulation directly reflects the varying degree of loners' externality that we introduced conceptually. In each case, the frequency of cooperators within groups becomes $\frac{r_C}{r_C + r_D + \rho r_L}$ where $r_x$ denotes the number of participants choosing action $x$, as in Eq. (4). This manipulation was conducted within subjects, meaning that all participants experienced each of the three conditions in a randomized order. In each condition, we measured participants' action selections (C, D, or L) and beliefs about others' action selections ("How many others do you think will choose C, D, or L in this round?") in an incentivized manner. Refer to the github repository for more detailed protocols and the instruction slides (translated into English) used in the additional experiment.

The sample size was determined to match the main experiment. As this additional experiment was more exploratory, we did not pre-register any hypotheses.

## Statistical analyses

To evaluate the variability of statistical values of interest (e.g., cooperation rate, group success rate, and efficiencies), we primarily employed bootstrap simulations. Since we used a within-subject design with order randomized both within and across sessions, our data are primarily clustered by participant (and not by session). To account for this, we used participant as the unit of resampling and calculated all the pertinent values (such as the cooperation rate and its difference between the voluntary and mandatory conditions) from that resampled data, instead of repeating independent resampling for each value. The exact calculation of the bootstrapped 95% CIs for the main and the additional experiment proceeded as follows:

1. Resample $N$ (main experiment: 191; additional experiment: 182) individuals from the experimental sample of $N$ individuals with replacement. We count each individual sample as a new participant in the resampled data, even if multiple samples correspond to the same actual participant, to properly account for the variability among participants in the mixed-effects models.
2. Calculate the statistical values of interest (e.g., the difference in cooperation rates between the voluntary and mandatory conditions) from the resampled data.
3. Repeat Steps 1 and 2 for 1000 iterations to obtain the distribution of the estimation of the statistics.
4. Report the range between the 2.5 and the 97.5 percentile of the empirical distribution as the bootstrapped 95% CI of the focal statistics.

Error bars in the figures indicate the bootstrapped 95% CIs, with the center values corresponding to the original values from the experiment. We interpret a 95% CI not containing 0 as evidence that a statistically significant difference exists.

Additionally, we introduced regression models in the following analyses. First, to assess how the cooperation (or defection) decisions in the mandatory conditions and leaving decisions in the voluntary conditions correlate with each other via subjective belief about others' cooperativeness, we evaluated three logistic regression models, each with a random intercept for participant: (1) defection in the mandatory conditions (i.e., defect = 1, cooperate = 0) as the dependent variable, and subjective belief in the mandatory conditions and threshold values as independent variables (i.e., fixed effects); (2) leaving in the voluntary conditions as the dependent variable, and subjective belief in the *mandatory* conditions and threshold values as independent variables; (3) leaving in the voluntary conditions as the dependent variable, and defection in the mandatory conditions (i.e., defect = 1, cooperate = 0) and threshold values as independent variables. Second, to explore economic or psychological traits (measured separately from the main task) that can partly account for inclinations toward the individual option rather than collaboration, we built a mixed-effects logistic regression with a random intercept of participants consisting of the leaving decision in the voluntary conditions as the dependent variable, and risk aversion, inequity aversion, Interpersonal Reactivity Index, general trust, Intolerance of Uncertainty Scale, Cognitive Reflection Test scores, and threshold values as independent variables. See Supplementary Note 2 for detailed descriptions about each of these measures. Third, for the analysis of the additional experiment, we built another mixed-effects logistic regression with a random intercept of participants consisting of belief about the effective cooperation rate (Eq. (4)) as the dependent variable and loners' externality as an independent variable.

Last, we computed the area under the curve of the receiver operating characteristic (ROC-AUC) for each of the raw expectations about others' cooperation and the probability of their own cooperation being pivotal (which is calculated from raw expectation) in terms of the predictive power of action (see Supplementary Note 3 for details about the ROC analysis). Notice that when the threshold is 5, the ROC-AUC of the raw expectation and the pivotal probability coincide. This is because the pivotal probability increases monotonically as the raw expectation increases (i.e., the order does not change) when the threshold is 5, and ROC-AUC is a rank metric that solely depends on the order of the predictions. Therefore, we are only concerned with the threshold values of 2 and 4 in this comparison.

## Reporting summary

Further information on research design is available in the Nature Portfolio Reporting Summary linked to this article.

## Data availability

The data of our experiments are publicly available at https://github.com/ryutau/voluntary-collaboration[62]. There are no restrictions to accessing the data.

## Code availability

The code used in this study is available at the GitHub repository: https://github.com/ryutau/voluntary-collaboration[62]. The analyses were implemented in Python (v.3.10.11) using the matplotlib (v.3.7.2) library, numpy (v.1.24.4), pandas (v.2.0.3), pymer4 (v.0.8.0), scikit-learn (v.1.3.0), and scipy (v.1.12.0).

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

## Acknowledgements

We thank N. Mifune, N. Takahashi, and A. Ueshima for helpful comments on a previous draft, Y. Shimodaira for support in experimental preparation,

and A. Todd for editing the manuscript. We are also grateful for support from the Center for Experimental Social Sciences at Hokkaido University, the Experimental Economics Lab at Meiji Gakuin University, the UTokyo Center for Integrative Science of Human Behavior, and the Joint Usage/ Research Center at ISER. This study was supported by the Japan Society for the Promotion of Science (grant no. JP16H06324 to T.K., no. JP23KJ0781 to R.M., and JP20H05631 and JP 23H00055 to N.H.), the Japan Science and Technology Agency CREST [grant no. JPMJCR17A4 (17941861)] to T.K., the Japanese Society of Social Psychology to R.M., and the Foundation for the Fusion of Science and Technology to R.M.

## Author contributions

Conceptualization: R.M., N.H., T.K. Methodology: R.M., N.H., T.K. Investigation: R.M. Visualization: R.M. Supervision: T.K. Writing—original draft: R.M., T.K. Writing—review & editing: R.M., N.H., T.K.

## Competing interests

The authors declare no competing interests.
