## [Peer Review File · Nature Communications]

An outside individual option increases optimism and facilitates collaboration when groups form flexiblyReviewers' Comments:

Reviewer #1:

Remarks to the Author:

In the present paper, the authors investigate cooperation rates in a voluntary step-level public goods game under varying levels of required coordination and compare it to a mandatory step-level public goods game. In line with their theoretical results, the authors find that successful coordination increases when people can voluntarily join groups and they provide some evidence for the reasons (namely, self-selection of cooperators as well as increased optimistic beliefs).

I read the paper with much interest. The question is very interesting and timely. Moreover, the study is set up very elegantly. The model is simple and, yet, provides some rich implications and the empirical test closely follows the model.

As such, I am very confident that this paper can make an important contribution to the cooperation literature, providing a general model on the role of voluntary participation in group cooperation and coordination that could inspire a lot of future work (e.g., extending the work with regards to group sizes, benefits of being a loner, manipulating the externality etc.). The writing is also very clear and the figures are very instructive and helpful!

Overall, I highly recommend publication. Having said that, I do have a couple of comments on the interpretation of the general effects that were a bit puzzling to me.

(1) One of the main "take-aways" from the study seem to be that voluntary participation in a step-level PG increases cooperation (especially when more coordination is required). Yet, there is an important distinction between "group success" (whether a group manages to coordinate successfully) and "cooperation rate" and this, seemed to sometimes blurred in the writing (e.g., p. 3: "voluntary participation increased the rate of individual cooperation within groups, up to the point where most groups succeeded in producing collective benefits.").

When looking at Figure S2, comparing the proportion of cooperators (orange part of the pie chart), it actually looks like the opposite: There is a consistently higher frequency of choosing the cooperative action in the mandatory treatment. In Figure 2A, however, it looks exactly like the opposite is true. I assume Figure 2A shows the cooperation rate for those who opted to join a group and removes the loners.

So, if I am not mistaken, the story is a bit more complicated: Forcing people to stay (Mandatory treatment) actually seem to increase cooperation rates. Yet, because there is still a lot of free-riding, groups often do not succeed, whereas the self-selection mechanism in the Voluntary treatment helps to create groups with a higher likelihood of "committed cooperators" and failure rates decrease.

Again, if I do not get something wrong here, I think this should be reported more transparently. If the point would be to just increase overall cooperation levels, 'forcing people into groups' would actually be a better choice / institution.

(2) It was not entirely clear to me how groups were formed. Since it was one-shot and groups formed based on the basis of who opted in and who opted out, many different groups could be formed. For example, if 10 people opted to stay, there are many different ways how two groups of five can be formed among these 10. In some combinations groups may be more or less successful. Hence, I was wondering how the data in Fig 2B or C was calculated. Did the authors calculate all possible group combinations among the participants who opted into the public good?

(3) Related to (1); Also in Fig 1C, the best response depending on beliefs allows for a larger proportion of choosing cooperation under Mandatory vs. Voluntary. From this perspective, let's assume for a moment that beliefs are uniformly distributed across people. From this perspective, one would again expect that a higher proportion of the population would cooperate under Mandatory compared to Voluntary (albeit less successfully).

This is of course no problem, but if I interpret this correctly, the story is rather: Voluntary entering in

this context increases group success but some people who would have cooperated actually become "loners" under Voluntary. For example, in Fig. S2, 41% of participants defect and 58% leave (under threshold 5). Similarly, 56% defect and 93% leave (under threshold 4).

Hence, optional participation seems to not only increase group success but also crowd out some people who would have cooperated under "Mandatory". So essentially, there seems to be trade-off: Voluntary participation increases group success (only those willing to cooperate and confident that others will do enter) but also some cooperators (likely with too pessimistic beliefs) that would have cooperated under "Mandatory" become loners (as also shown in Fig. 3B).

I feel like this should be discussed more. Figure S2 is quite instructive in this regard (especially the pie-charts). The authors should consider moving that to the main-text.

(4) Having a version of the game with a threshold of 5 is, of course, nice as a benchmark. But, from my understanding, there are no free-riding incentives in this case anymore and, hence, this is not a social dilemma but a pure coordination game. There is nothing to gain from "free-riding" in this scenario. Maybe this could be highlighted a bit more as certain psychological motives (e.g., not being taken advantage of or trying to free-ride on others) should not play a role in this setting anymore. Talking about "cooperation" or "defection" in this setup (i.e., threshold of 5) could be seen as a bit misleading.

(5) Regarding the model; I was wondering if second-order beliefs should also play a role here. Based on Fig S1, it seems like the mechanism of the model is that voluntary participation increases cooperation rates based on the assumption that some will become loners and that the highest proportion of these loners would be defectors if they would be forced to stay (see also Fig. 1C). Yet, what if defectors (those who truly want to take advantage of others) can anticipate this dynamic and therefore opt-in? Such second-order (or higher-order) belief dynamics could be at least discussed as a potential future outlook (if I am correct here and they are, indeed, not considered in the current model).

(6) Under threshold of 2, the best group outcome is if two people cooperate and three people defect. Cooperation above 2 becomes inefficient, since it is a step-level game. As such, groups also need to find a balance between efficient coordination (only two cooperators) and fairness (three people take advantage of cooperation in the most efficient case). The authors do not consider this, but people may also opt out of the game in this situation due to aversion to inequality. If everyone opts out, overall earnings may be lower but at least everyone has the same level of wealth. Hence, some people may opt out of the game not due to beliefs but because they anticipate some unfairness (in terms of earnings distribution).

(7) Lastly (but actually quite important), I was wondering about the robustness or exact interpretation of one finding mentioned prominently in the abstract: "... ensuring that the existence of an individual option robustly aids collaborative success by fostering cooperation through improved optimism within groups" ("second pathway" in the results). In Figure 3D for threshold 2 and 4 (which still has elements of a social dilemma – whereas threshold 5 is a pure coordination game) the proportion of orange dots seem quite similar to the blue dots and the most dots seem to be gray. Furthermore, as mentioned above, many cooperating participants may have opted to leave the PG (which is not considered in this figure). Yet, the analyses only consider those who stayed. Hence, the self-selection effect seems to be confounded with beliefs here. If beliefs increase the chance to stay, shouldn't that imply that those who stay have more optimistic beliefs by definition? Hence, I am not quite seeing the point that voluntary participation makes the population more optimistic but rather that those who stay are more optimistic (which is simply an effect of self-selection). Maybe this just subtle but it could be misinterpreted as "selective play" has an overall effect on beliefs.

Reviewer #2:

Remarks to the Author:

This paper studies voluntary participation in public good games. It presents a model, as well as experimental results to highlight the fact that voluntary participation can lead to cooperation in public good games.

The model assumes a threshold public goods game, where the benefits flow to the participants once the number of cooperators meet a predetermined threshold. If the option to abstain exists, then a payoff is guaranteed for those that do not participate in the PGG. The model starts by assuming that players hold beliefs about whether others will cooperate γ , and computes the expected payoff as a function of this parameter assuming agents best-respond to their own beliefs. Further to this, the proportion of cooperation in a population of players can be computed by assuming a distribution ϕ of subjective beliefs in the population. Then, the level of cooperation with and without the option to abstain can be computed for different distributions ϕ .

The experiments basically find that (in alignment with the model), the option to abstain increases cooperation. An interesting feature of the experiments is that they can explore how beliefs change for the voluntary and non-voluntary PGG. The authors focus on whether "pessimistic" defectors are filtered out of voluntary games, or change their beliefs in the presence of abstention in order to become cooperators: they find that the mechanism is the former unless the stakes are low.

The authors then embark on reconciling the fact that Gross et al. [ref 26] find negative effects of voluntary participation. I don't find this particularly interesting, given that the set up in that article seems to be such that those that abstain can still benefit from other's cooperation. Thus, I believe this tension is non-existent and the discussion adds little. Moreover, whether the option to abstain is meaningful depends on the nature of the public good, and in my opinion has nothing to do with local or global games.

The main issue that I have with this paper is that it treats voluntary participation as something new. In terms of modelling, this has been extensively studied before: see particularly the work of Hauert and Brandt, which is now more than 20 years old. The findings are very similar, and there is no mention or comparison here.

Another issue with the paper is that the model is not general enough. It only considers a small number of possible thresholds, and restricts itself to numerical results in a very small range of parameter. Some generality may make this contribution stronger.

The paper also doesn't consider dynamics explicitly, which is important in order to endogenise ϕ . Altogether, I find the contribution marginal and better suited to a specialised journal once the appropriate comparisons with the work of Hauert et al, and Brandt et al. are considered.

- References:

- Hauert C, De Monte S, Hofbauer J, Sigmund K. Volunteering as red queen mechanism for cooperation in public goods games. *Science*. 2002 May 10;296(5570):1129-32.
- Brandt, Hannelore, Christoph Hauert, and Karl Sigmund. "Punishing and abstaining for public goods." *Proceedings of the National Academy of Sciences of the United States of America* 103.2 (2006): 495.
- Hauert C, De Monte S, Hofbauer J, Sigmund K. Replicator dynamics for optional public good games. *Journal of Theoretical Biology*. 2002 Sep 21;218(2):187-94.
- Hauert C, Traulsen A, Brandt H, Nowak MA, Sigmund K. Via freedom to coercion: the emergence of costly punishment. *science*. 2007 Jun 29;316(5833):1905-7.

Reviewer #3:

Remarks to the Author:

I thought this was a terrific paper. The theorizing is lucid, the methods are rigorous, and the results are compelling. Although the idea of opting in vs. out has been examined elsewhere, I really like the analysis of individual beliefs.

That said, I have several questions about loners' externality. First, given that this is perhaps the most novel insight (as italicized in the frontend of the paper, line 40), I am slightly disappointed that it was not examined more directly in the main experiment. Although I appreciate the theoretical discussion on p. 11-13, I would have liked to see, for instance, how loners' externality changes people's beliefs about group cooperation (y). Short of adding a new study, perhaps you could move the theoretical discussion to the top of the Results section and use it to motivate your experimental design (as opposed to ending the paper with it).

Second, I am curious about how to conceptualize loners' externality as a continuous variable. If global versus local public goods games represent the extreme ends of the continuum, how should we think about intermediate cases? When do we see $p=.05$ such that implementing individual options do not make any difference?

Third, I am a bit confused about the terms global and local. In the global case, public goods games are still played at the "local" level of predefined groups. Would it make sense to just call these cases opting in vs. out?

Line 242: This is a nice, punchy line, but it should say "Voluntary participation does just that."

Reviewer 1

R.1.0 In the present paper, the authors investigate cooperation rates in a voluntary step-level public goods game under varying levels of required coordination and compare it to a mandatory step-level public goods game. In line with their theoretical results, the authors find that successful coordination increases when people can voluntarily join groups and they provide some evidence for the reasons (namely, self-selection of cooperators as well as increased optimistic beliefs).

I read the paper with much interest. The question is very interesting and timely. Moreover, the study is set up very elegantly. The model is simple and, yet, provides some rich implications and the empirical test closely follows the model.

As such, I am very confident that this paper can make an important contribution to the cooperation literature, providing a general model on the role of voluntary participation in group cooperation and coordination that could inspire a lot of future work (e.g., extending the work with regards to group sizes, benefits of being a loner, manipulating the externality etc.). The writing is also very clear and the figures are very instructive and helpful!

Overall, I highly recommend publication. Having said that, I do have a couple of comments on the interpretation of the general effects that were a bit puzzling to me.

Response: We thank the reviewer for supporting the publication of our manuscript and providing constructive feedback. We carefully revised the manuscript in light of your comments and believe that this has greatly improved the manuscript. Below, we give a point-by-point response to each comment. For any line numbers from the main text cited in this document, please refer to the corresponding lines in the two-column version of the revised main manuscript file with all tracked changes accepted.

R.1.1 (1) One of the main “take-aways” from the study seem to be that voluntary participation in a step-level PG increases cooperation (especially when more coordination is required). Yet, there is an important distinction between “group success” (whether a group manages to coordinate successfully) and “cooperation rate” and this, seemed to sometimes blurred in the writing (e.g., p. 3: “voluntary participation increased the rate of individual cooperation within groups, up to the point where most groups succeeded in producing collective benefits.”).

When looking at Figure S2, comparing the proportion of cooperators (orange part of the pie chart), it actually looks like the opposite: There is a consistently higher frequency of choosing the cooperative action in the mandatory treatment. In Figure 2A, however, it looks exactly like the opposite is true. I assume Figure 2A shows the cooperation rate for those who opted to join a group and removes the loners.

So, if I am not mistaken, the story is a bit more complicated: Forcing people to stay (Mandatory treatment) actually seem to increase cooperation rates. Yet, because there is still a lot of free-riding, groups often do not succeed, whereas the self-selection mechanism in the Voluntary treatment helps to create groups with a higher likelihood of “committed cooperators” and failure rates decrease.

Again, if I do not get something wrong here, I think this should be reported more transparently. If the point would be to just increase overall cooperation levels, ‘forcing people into groups’ would actually be a better choice / institution.

Response: Thank you very much for highlighting these important distinctions. You are right to differentiate between “group success” and “individual cooperation” in collaborative settings modeled as the threshold public goods game. When the loner option is available, we need to further distinguish between two types of (individual) cooperation rates: (A) the proportion of cooperators in the entire population (i.e., $\#C/(\#C+\#D+\#L)$ or the number of cooperators) and (B) the proportion of cooperators among those who opt in (i.e., excluding loners; $\#C/(\#C+\#D)$).

In our paper, we consistently focus on the latter, B-type definition of cooperation rate (excluding the number of loners from the denominator) until the middle of the Results section (covering theoretical predictions and experimental results, including Fig. 2A), as it is the very quantity that determines group success in our setting. Because our motivation is to understand the determinants of collaborative success, we believe that the rate that directly impacts the group outcome should be highlighted (instead of calling the sheer number of cooperators the “cooperation rate”).

To clarify the important distinction in defining cooperation rate, we have revised the Results in

the main text as follows.

(Lines 336 to 344) *“Figure 2B displays participants’ cooperation rate for the public good within groups in each condition. Note that here, as defined in Eq. 3, cooperation rate refers specifically to the frequency of cooperators among those who choose to participate in groups $[\#C/(\#C+\#D)]$ rather than the frequency among the entire population $[\#C/(\#C+\#D+\#L)]$, as it is the former frequency that determines the outcome of group collaboration.”*

We have also revised the legend of Fig. 2B.

(Fig. 2 legend) *“Voluntary participation supports collaborative success. ... (B) Individual cooperation rate $[\#C/(\#C+\#D)]$, (C) group success rate, and (D) normalized (i.e., ratio against the highest possible payoff for each threshold) average payoff in each condition.”*

Having made these distinctions, as you correctly pointed out, introducing an outside individual option gives rise to a trade-off between the decrease in the exact number (count) of cooperators ($\#C$; the type-A cooperation rate above) and the increase in the frequency of cooperators among those participating in groups ($\#C/(\#C+\#D)$; the type-B cooperation rate). We do agree with you that it is of particular importance to precisely interpret the consequences of this trade-off. In the last part of the Results section (please see Lines 530 to 686 in the subsection titled Theoretical mapping), we thus have discussed (and have explored its empirical implication in a new experiment) how the type of cooperation rate that is most relevant to group success is determined by the nature of the goods or group in question (please see also R.1.3).

R.1.2 (2) It was not entirely clear to me how groups were formed. Since it was one-shot and groups formed based on the basis of who opted in and who opted out, many different groups could be formed. For example, if 10 people opted to stay, there are many different ways how two groups of five can be formed among these 10. In some combinations groups may be more or less successful. Hence, I was wondering how the data in Fig 2B or C was calculated. Did the authors calculate all possible group combinations among the participants who opted into the public good?

Response: We apologize for the lack of clarity in our explanation of how we calculated the group success rate (Fig. 2B) from the empirical distribution of participants’ actions. As you pointed out, there are many different ways in which groups are formed. To address this, we computed the average group success rate for potential group formations using binomial expansions (assuming a sampling with replacement): $p_{\text{success}} = \sum_{k=q}^5 \binom{5}{k} p_{\text{coop}}^k (1 - p_{\text{coop}})^{5-k}$. We have included this equation defining the

group success rate in the revised manuscript.

(Lines 358 to 362) *“One important measure is how likely groups are to succeed in generating the collective benefits (“group success rate”; probability of a group gathering enough cooperators; $p_{success} = \sum_{k=q}^5 \binom{5}{k} p_{coop}^k (1 - p_{coop})^{5-k}$).”*

R.1.3 (3) Related to (1); Also in Fig 1C, the best response depending on beliefs allows for a larger proportion of choosing cooperation under Mandatory vs. Voluntary. From this perspective, let's assume for a moment that beliefs are uniformly distributed across people. From this perspective, one would again expect that a higher proportion of the population would cooperate under Mandatory compared to Voluntary (albeit less successfully).

This is of course no problem, but if I interpret this correctly, the story is rather: Voluntary entering in this context increases group success but some people who would have cooperated actually become “loners” under Voluntary. For example, in Fig. S2, 41% of participants defect and 58% leave (under threshold 5). Similarly, 56% defect and 93% leave (under threshold 4).

Hence, optional participation seems to not only increase group success but also crowd out some people who would have cooperated under “Mandatory”. So essentially, there seems to be a trade-off: Voluntary participation increases group success (only those willing to cooperate and confident that others will do enter) but also some cooperators (likely with too pessimistic beliefs) that would have cooperated under “Mandatory” become loners (as also shown in Fig. 3B).

I feel like this should be discussed more. Figure S2 is quite instructive in this regard (especially the pie-charts). The authors should consider moving that to the main-text.

Response: Thank you for highlighting this key issue. As you have correctly pointed out, voluntary participation (entering) in this context increases group success but also decreases the number of cooperators (#C) as some potential cooperators choose to become “loners.” This potential decrease in the “number” of cooperators is indeed predicted by our model under certain belief distributions (such as the uniform distribution in your example) and was also observed in our data. Upon your suggestion, in the revised manuscript, we have explicitly mentioned this point when we provide the model in the Results.

(Lines 190 to 202) *“As seen in the green intervals in Fig. 1C: Voluntary, leaving is the best possible action across the thresholds unless the player's γ is sufficiently high. In other words, those who hold low expectations about their peers' cooperation will naturally choose the individual option outside the group. These loners will cover “pessimistic defectors” who would*

not have cooperated within groups out of fear under mandatory participation and may also include some potential cooperators who hold relatively low expectations. As such, voluntary participation will work as a self-selection mechanism that filters pessimistic individuals out of collaborative efforts.”

Also, we agree that the pie charts in Supplementary Fig. 2 in the original manuscript are instructive in showing the exact number of participants choosing each action in each condition. Thus, as suggested, we have included them as Fig. 2A in the revised main text.

(Fig. 2 legend) *“Fig. 2. Voluntary participation supports collaborative success. (A) Breakdown of the participants choosing each action in each condition. The pie charts depict the number of participants who selected each action (orange: cooperate [C], blue: defect [D], and green: leave [L]), with exact numbers in the wedges.”*

(Lines 332 to 336) *“Figure 2A shows the exact number of participants choosing each action in each condition. Responding to the difference in the threshold values and whether participation was mandatory or voluntary, participants indeed changed their action selections.”*

Despite this trade-off, we maintain that the overall impact of voluntary participation is positive for three reasons. First, as detailed in our response to R.1.1, the central focus is not on the sheer number of cooperators, but on whether the groups can successfully launch collaboration. In our setting, it is the ratio of cooperators to defectors that determines the group’s success.

Second, regarding population-level efficiency (i.e., average payoff), our results (Fig. 2C) show that the efficiency is generally higher under voluntary conditions (except for $q = 2$, where most mandatory groups were already successful), yielding a much higher average payoff for their members (30 or 40 points) compared to the loner option (20 points).

Last, in our framework, the collaboration corresponds to a potential solution that can improve participants’ payoffs if successful, albeit with inherent uncertainties due to mutual self-interested temptations. Opting out here is not a lifetime isolation from the group but rather a choice of safer individual activities over a specific opportunity for collaboration.

For these reasons, acknowledging the importance of trade-off, we believe that the problem is mostly circumvented in our setting.

R.1.4 (4) Having a version of the game with a threshold of 5 is, of course, nice as a benchmark. But, from my understanding, there are no free-riding incentives in this case anymore and, hence, this is not a social dilemma but a pure coordination game. There is nothing to gain from “free-riding” in this scenario. Maybe this could be highlighted a bit more as certain psychological motives (e.g., not being taken advantage of or trying to free-ride on others) should not play a role in this setting anymore.

Talking about “cooperation” or “defection” in this setup (i.e., threshold of 5) could be seen as a bit misleading.

Response: Appreciating the reviewer’s concern, we would like to note that in some key literature on the stag hunt game, which is a two-person instance of a pure coordination game, the term “cooperation” has been used similarly to refer to the collaborative action (e.g., Skyrms, 2001). However, as you correctly pointed out, we acknowledge the importance of highlighting that a threshold value of 5, which equals the group size, effectively removes the free-riding temptation. Thus, we have included multiple sentences to clarify this point in the Results section of our revised manuscript.

(Lines 134 to 136) *“Here, we consider three threshold values: $q = 2, 4, \text{ or } 5$ (out of 5), representing a strong, weak, or null temptation to free ride, respectively.”*

(Lines 183 to 187) *“Note that with the threshold of 5 (Fig. 1C, right), greedy defection cannot exist, as free riding is no longer possible (i.e., a player’s own defection guarantees failed group endeavor), effectively transforming the game into a pure coordination game.”*

R.1.5 (5) Regarding the model; I was wondering if second-order beliefs should also play a role here. Based on Fig S1, it seems like the mechanism of the model is that voluntary participation increases cooperation rates based on the assumption that some will become loners and that the highest proportion of these loners would be defectors if they would be forced to stay (see also Fig. 1C). Yet, what if defectors (those who truly want to take advantage of others) can anticipate this dynamic and therefore opt-in? Such second-order (or higher-order) belief dynamics could be at least discussed as a potential future outlook (if I am correct here and they are, indeed, not considered in the current model).

Response: Thank you very much for discussing this important direction. We agree that second-order beliefs, or even higher-order beliefs, can matter in this context. Indeed, in the original manuscript, we briefly touched on the idea of players engaging in recursive updating of beliefs, fluctuating between optimism and pessimism, rather than merely holding beliefs intact across conditions.

However, we realized that fully developing the model in this direction extends beyond the scope of our current research. We think that the results of recursive belief updating in threshold public goods

game would heavily depend on additional individual factors such as the depth of thought (i.e., the number of times they iterate in thinking) and their subjective perceptions of other players' decision-making noise. Furthermore, as these factors are known to vary between individuals (e.g., Nagel, 1995; Coricelli & Nagel, 2009), we would further need to assume their distributions among the population. Given these complexities, in the original manuscript, we discussed only the equilibrium case where beliefs converge, instead of directly dealing with the fluctuation of beliefs (please see Lines 242 to 291 and Supplementary Information 1).

Nevertheless, we agree with you that fully endogenizing players' beliefs with some computational modeling is an interesting possibility. Therefore, we have included this perspective in the Discussion of our revised manuscript, offering it as a potential direction for future studies (see also our response to R.2.3).

(Lines 826 to 841) *“Second, our model does not explicitly predict how individuals form their subjective beliefs. We first assume that these beliefs remain intact regardless of whether the group participation is voluntary or mandatory (Fig. 1B & C), and then examine equilibrium prediction, where beliefs ultimately converge with the distribution of players' actions (Supplementary Information 1). We took this approach because we found it intractable to explicitly model belief formation given that it depends heavily on additional individual factors. These factors include players' depth of thought and their perceptions of others' decision-making noise—aspects that are known to vary widely among individuals and contexts⁵⁹⁻⁶¹. Nevertheless, we acknowledge the potential value and interest in fully endogenizing players' beliefs.”*

R.1.6 (6) Under threshold of 2, the best group outcome is if two people cooperate and three people defect. Cooperation above 2 becomes inefficient, since it is a step-level game. As such, groups also need to find a balance between efficient coordination (only two cooperators) and fairness (three people take advantage of cooperation in the most efficient case). The authors do not consider this, but people may also opt out of the game in this situation due to aversion to inequality. If everyone opts out, overall earnings may be lower but at least everyone has the same level of wealth. Hence, some people may opt out of the game not due to beliefs but because they anticipate some unfairness (in terms of earnings distribution).

Response: As you correctly pointed out, an increased number of cooperators within a group does not always lead to better outcomes in threshold public goods games, because any cooperation beyond the threshold does not contribute additional gains to the group and rather decreases efficiency by the incurred costs of cooperation. This caveat of “overcooperation” becomes more prominent when the

threshold is lower (e.g., $q = 2$). In the original manuscript, we explicitly mentioned this point when presenting the efficiency data (cf. Fig. 3D).

(Lines 377 to 383) *“A rise in cooperation rates within the voluntary groups does not necessarily result in greater efficiency for two distinct reasons: overcooperation within groups and overpresence of loners outside of groups. Overcooperation could reduce the average payoff because contributions above the threshold do not generate any further gain for the group.”*

We acknowledge your point that, because of these caveats, players might be motivated by factors other than individual-payoff maximization, such as efficiency consideration or aversion to unfairness. Interestingly, especially under lower threshold values, these different motives are mutually incompatible as long as one remains in a group and thus might prompt the decision to opt out. If this were a major factor, we would expect to observe that a substantial number of participants with optimistic beliefs will nevertheless abstain from participating in groups, leading to a greater fluctuation in the leaving rate as a function of beliefs under lower threshold values. However, our data in Fig. 3A (green) and Supplementary Fig. 2 did not exhibit such patterns. Additionally, as reported in the (revised) Supplementary Information 2 and Supplementary Table 1, the analyses of correlation between participants' individualistic choices and their economic and psychological characteristics measured by additional tasks indicate that there was no significant correlation between the choice to opt out from groups and the degree of inequity aversion (both advantageous and disadvantageous, as measured with the Fehr-Schmidt utility model). Moreover, the overall rate of participants opting out was greater when the threshold value was 4 as compared to 2, suggesting the primary factor inducing the decision to opt out in our experiment was the pessimistic anticipation of collaborative failure, which can be explained better by individual-payoff maximization than by the other motivations.

That being said, we recognize the relevance of these motivational nuances in broader collaborative contexts. Given that our study collected only a single data point per participant for each condition, it remains challenging to fully unravel the subtle variations in their motivations. Therefore, we deem it a promising direction for future research to further investigate these motivational aspects, possibly through different experimental designs aimed specifically at identifying what quantity individuals seek to maximize.

(Lines 813 to 825) *“It may also be plausible that human individuals are motivated by factors other than payoff maximization^{8–10,35}, such as efficiency and fairness concerns (but see also Supplementary Information 2 and Supplementary Table 1 for the irrelevance of fairness concerns in our main experiment). We believe that our game incorporates the minimal incentive structures underlying collaborative situations and thus provides benchmark predictions of what can happen under voluntary and mandatory participation, as well as a solid starting point for future investigations of other possible cognitive and motivational*

aspects.”

R.1.7 (7) Lastly (but actually quite important), I was wondering about the robustness or exact interpretation of one finding mentioned prominently in the abstract: “... ensuring that the existence of an individual option robustly aids collaborative success by fostering cooperation through improved optimism within groups” (“second pathway” in the results). In Figure 3D for threshold 2 and 4 (which still has elements of a social dilemma – whereas threshold 5 is a pure coordination game) the proportion of orange dots seem quite similar to the blue dots and the most dots seem to be gray. Furthermore, as mentioned above, many cooperating participants may have opted to leave the PG (which is not considered in this figure). Yet, the analyses only consider those who stayed. Hence, the self-selection effect seems to be confounded with beliefs here. If beliefs increase the chance to stay, shouldn’t that imply that those who stay have more optimistic beliefs by definition? Hence, I am not quite seeing the point that voluntary participation makes the population more optimistic but rather that those who stay are more optimistic (which is simply an effect of self-selection). Maybe this just subtle but it could be misinterpreted as “selective play” has an overall effect on beliefs.

Response: Thank you very much for clarifying this important point. First, the phrase “improved optimism within groups” was intended to encompass both pathways under voluntary participation: (a) the self-selection of optimists and (b) the optimistic update among (some) individuals who keep participating in groups under mandatory as well as voluntary conditions. These pathways collectively contribute to forming groups with a higher level of optimism under voluntary conditions than mandatory ones. We have revised the wording to make this explicit in the Abstract and the Results.

(Abstract) *“First, we find that voluntariness in collaboration increases the likelihood of group success via two pathways, both contributing to form more optimistic groups: pessimistic defectors are filtered out from groups, and some individuals update their beliefs to become cooperative.”*

(Lines 75 to 83) *“Further, two distinct behavioral mechanisms—pessimistic defectors being filtered out from groups (i.e., self-selection of optimists into groups)³⁷ and some pessimists switching from defection to cooperation via improved subjective beliefs about others’ cooperativeness—together operate under voluntary participation, contributing to form groups with a higher level of optimism and increasing the frequency of cooperators within groups.”*

More importantly, we acknowledge the need for precise verification of the second pathway we

proposed. To clarify, our claim regarding the second pathway is not that the entire population becomes more optimistic under voluntary participation compared to mandatory participation. Instead, we suggest that under voluntary conditions, some individuals shift from defection to cooperation (more so than the opposite shift), accompanied by an optimistic updating of beliefs, thereby increasing the overall cooperation rate beyond what is explained by self-selection (i.e., defectors are more likely to opt out than cooperators) alone. We realize that our original description, especially the aggregate-level belief-change analysis (*t* testing), may have been unclear in this regard, so we have decided to revise the paragraph with the following two steps.

We believe that a straightforward way to verify our claim is to limit the scope of the investigation to “non-loners”, those who participate in groups under both mandatory (M) and voluntary (V) conditions; here, as those who leave the group under V conditions are removed from the analysis, the direct effect of self-selection (i.e., differential leaving rate) on cooperation rate is removed. Among non-loners, we have thus examined (1) whether the cooperation rate is still higher in V compared to M conditions, and if so, (2) whether those action changes (from defect in M to cooperate in V; or from cooperate in M to defect in V) correlate with changes in their beliefs within individuals.

Fig. 3C directly corresponds to the first point, confirming that non-loners’ cooperation rate is higher under the voluntary condition than the mandatory condition. Although a visual inspection of Fig. 3D might give the impression that the numbers of orange and blue dots are not distinguishable in each panel, direct statistical tests yield significantly positive differences under $q = 4, 5$, and an insignificant but mostly positive difference in $q = 2$ (95% CI $[-0.02, 0.19]$). Besides this overall pattern, Fig. 3D was designed to visually represent how the action changes and the belief changes correlate among non-loners within each individual (addressing the second point directly). It is certainly true that those who remain in groups have more optimistic beliefs almost by definition in the between-individuals comparison (Fig. 3C), but the point here is the within-individual belief changes. In Fig. 3D, each dot represents a participant, with its position indicating the within-individual belief change and its color indicating the within-individual action change. The visual inspection here allows us to see whether the dots’ color and position correlate, in such a way that orange dots (representing from defecting in M to cooperating in V) are primarily located above and to the left of the diagonal (indicating becoming more optimistic in V conditions). We have also included a direct statistical test confirming the correlation between action changes and belief changes within individuals.

Accordingly, we have revised the paragraph presenting the results concerning the second pathway as follows:

(L464 to 509) “Next, we consider the second pathway. Can voluntary participation lead some pessimistic defectors to form more optimistic beliefs and turn to cooperation within groups, rather than merely prompting them to opt out from groups? To examine this point, we now focus on participants who stayed in groups under both voluntary and mandatory conditions in

each threshold (i.e., non-loners). Removing the loners, the cooperation rates within groups were still significantly higher in the voluntary condition than in the mandatory condition under the threshold values of 4 and 5 (Fig. 3C; the difference in cooperation rates among non-loners: $q = 4$: 0.09, 95% CI [0.01, 0.18]; $q = 5$: 0.14, 95% CI [0.09, 0.20]). When the threshold is 2, the difference was mostly positive but contains zero ($q = 2$: 0.08, 95% CI [-0.02, 0.19]). These results suggest that there was a net cooperative action shift among those who kept opting in to groups; participants were more likely to switch from defection in the mandatory condition to cooperation in the voluntary condition than the opposite.

Then, did this cooperative action shift parallel the optimistic updating of beliefs, as we conjectured? Figure 3D displays scatter plots of each participant's subjective beliefs about others' cooperativeness, γ , in the mandatory (x axis) and the voluntary (y axis) conditions. Here, each dot represents a non-loner who stayed in groups, with its position indicating the belief change and its color indicating the action change (see the caption of Fig. 3D for details). As seen in the figure, for each threshold value, individuals who showed positive (cooperative) action change (orange) were primarily distributed above the diagonal, becoming more optimistic in the voluntary condition. The relationship between the within-individual changes in action and belief was statistically significant (mixed-effects regression: $z = 9.12$, $\beta = 0.82$, $p < .001$, 95% CI [0.65, 1.00]), confirming that the cooperative action changes from the mandatory to voluntary conditions were accompanied by optimistic belief changes at the individual level. Together, these results suggest that, beyond the effect of self-selection (filtering out pessimistic defectors), voluntary participation encourages a significant number of individuals to develop optimism regarding others' cooperativeness and turn to cooperation within groups."

References:

- Skyrms, B. The stag hunt. *Proc. Addresses Am. Philos. Assoc.* **75**, 31–41 (2001).
- Nagel, R. Unraveling in guessing games: An experimental study. *Am. Econ. Rev.* **85**, 1313–1326 (1995).
- Coricelli, G. & Nagel, R. Neural correlates of depth of strategic reasoning in medial prefrontal cortex. *Proc. Natl. Acad. Sci. U. S. A.* **106**, 9163–9168 (2009).

Reviewer 2

R.2.0 This paper studies voluntary participation in public good games. It presents a model, as well as experimental results to highlight the fact that voluntary participation can lead to cooperation in public good games.

The model assumes a threshold public goods game, where the benefits flow to the participants once the number of cooperators meet a predetermined threshold. If the option to abstain exists, then a payoff is guaranteed for those that do not participate in the PGG. The model starts by assuming that players hold beliefs about whether others will cooperate γ , and computes the expected payoff as a function of this parameter assuming agents best-respond to their own beliefs. Further to this, the proportion of cooperation in a population of players can be computed by assuming a distribution ϕ of subjective beliefs in the population. Then, the level of cooperation with and without the option to abstain can be computed for different distributions ϕ .

The experiments basically find that (in alignment with the model), the option to abstain increases cooperation. An interesting feature of the experiments is that they can explore how beliefs change for the voluntary and non-voluntary PGG. The authors focus on whether "pessimistic" defector are filtered out of voluntary games, or change their beliefs in the presence of abstention in order to become cooperator: they find that the mechanism is the former unless the stakes are low.

The authors then embark on reconciling the fact that Gross et al. [ref 26] find negative effects of voluntary participation. I don't find this particularly interesting, given that the set up in that article seems to be such that those that abstain can still benefit from other's cooperation. Thus, I believe this tension is non-existent and the discussion adds little. Moreover, whether the option to abstain is meaningful depends on the nature of the public good, and in my opinion has nothing to do with local or global games.

Response: We thank the reviewer for the thorough review of our manuscript and the constructive feedback. We carefully revised the manuscript in light of your comments and believe that this has improved the manuscript.

Before addressing the three main issues raised by the reviewer point-by-point (please see R.2.1 through R.2.3), we wish to discuss two points in the above comment.

First, the reviewer summarized the mechanisms facilitating collaborative success under voluntary participation as: "The authors focus on whether 'pessimistic' defector are filtered out of voluntary games, or change their beliefs in the presence of abstention in order to become cooperator: they find that the mechanism is the former unless the stakes are low." However, we respectfully argue that our

empirical data provide evidence for the existence of the second mechanism as well. As we have explained in detail in our reply to Reviewer 1's Comment 7 (R.1.7), even if we focus on the cooperation rate among only those who consistently belonged to groups under the voluntary condition as well as under the mandatory condition ("non-loners"), there are still significantly positive differences in cooperation rates between the two conditions under $q = 4, 5$ ($q = 4$: 95% CI [0.01, 0.18]; $q = 5$: 95% CI [0.09, 0.20]), and an insignificant but mostly positive difference in $q = 2$ (95% CI [-0.02, 0.19]). This difference indicates that even after removing the direct effect of self-selection on cooperation (i.e., even after pessimistic defectors were filtered out of the games), the remaining participants behaved more cooperatively under the voluntary condition than under the mandatory condition (Fig. 3C). To shed light on how this difference emerged in relation to participants' belief changes between the mandatory and voluntary conditions, we then examined whether the individual action changes (i.e., from defect in M to cooperate in V or from cooperate in M to defect in V) correlate with changes in their subjective beliefs. Fig. 3D is specifically designed to inform on this point by visually representing the within-individual correlation between the action changes and the belief changes. In the figure, each dot represents a non-loner (the one who consistently belonged to groups across both conditions), with its position indicating the within-individual belief change and its color indicating the within-individual action change. The dot colors and positions indeed correlate, in such a way that orange dots (representing change from defect in M to cooperate in V) are primarily located above and to the left of the diagonal (indicating participants held more optimistic beliefs in V than in M). We have also executed a direct statistical test examining the correlation between action changes and belief changes within individuals. Thus, the second mechanism did work to promote greater cooperation in V conditions. As discussed in R.1.7, our presentation may not have been clear enough in the original manuscript, so we have now revised the paragraph verifying the second pathway as follows:

(L464 to 509) "Next, we consider the second pathway. Can voluntary participation lead some pessimistic defectors to form more optimistic beliefs and turn to cooperation within groups, rather than merely prompting them to opt out from groups? To examine this point, we now focus on participants who stayed in groups under both voluntary and mandatory conditions in each threshold (i.e., non-loners). Removing the loners, the cooperation rates within groups were still significantly higher in the voluntary condition than in the mandatory condition under the threshold values of 4 and 5 (Fig. 3C; the difference in cooperation rates among non-loners: $q = 4$: 0.09, 95% CI [0.01, 0.18]; $q = 5$: 0.14, 95% CI [0.09, 0.20]). When the threshold is 2, the difference was mostly positive but contains zero ($q = 2$: 0.08, 95% CI [-0.02, 0.19]). These results suggest that there was a net cooperative action shift among those who kept opting in to groups; participants were more likely to switch from defection in the mandatory condition to cooperation in the voluntary condition than the opposite.

Then, did this cooperative action shift parallel the optimistic updating of beliefs, as we conjectured? Figure 3D displays scatter plots of each participant's subjective beliefs about others' cooperativeness, γ , in the mandatory (x axis) and the voluntary (y axis) conditions. Here, each dot represents a non-loner who stayed in groups, with its position indicating the belief change and its color indicating the action change (see the caption of Fig. 3D for details). As seen in the figure, for each threshold value, individuals who showed positive (cooperative) action change (orange) were primarily distributed above the diagonal, becoming more optimistic in the voluntary condition. The relationship between the within-individual changes in action and belief was statistically significant (mixed-effects regression: $z = 9.12$, $\beta = 0.82$, $p < .001$, 95% CI [0.65, 1.00]), confirming that the cooperative action changes from the mandatory to voluntary conditions were accompanied by optimistic belief changes at the individual level. Together, these results suggest that, beyond the effect of self-selection (filtering out pessimistic defectors), voluntary participation encourages a significant number of individuals to develop optimism regarding others' cooperativeness and turn to cooperation within groups."

The second point concerns the comparison with Gross & De Dreu [new ref. 31] and our study. We agree with the reviewer that the impact of introducing the option to abstain depends on the nature of the public good, specifically, whether those who abstain are still within the scope of the good. When the loners are excluded from the good, the frequency of cooperators among those who choose to opt in effectively determines whether the group succeeds in producing the public good, whereas when the loners are not excluded, the frequency of cooperators in the entire population matters. In the original manuscript, we intended to refer to the former case as "local" and the latter as "global." In the revised manuscript, we have changed these terms to streamline the discussion (see also our response to R.3.3).

In contrast to the reviewer, however, we believe that these discussions are not trivial, especially given how the findings in Gross et al. have been digested and referred to in the subsequent literature. For example, Balliet and Lindström (2023; *Trends in Cognitive Science*) stated that "people are more willing to invest in a public good to solve a problem in the absence, than in the presence, of an option to solve the problem independently," and Gross et al. (2020; *Nature Communications*) emphasized that "self-reliance crowds out group cooperation," summarizing ref. 31. The findings from our study and theirs can be easily conflated if summarized as a generic question such as "whether the introduction of an outside individual option facilitates or hinders a collective option." We believe that explicitly clarifying the conditions where each conclusion holds adds value to the cooperation literature. Indeed, Reviewer 3 views the discussion about loners' externality as "perhaps the most novel insight" (R.3.1) in our paper.

R.2.1 The main issue that I have with this paper is that it treats voluntary participation as something new. In terms of modelling, this has been extensively studied before: see particularly the work of Hauert and Brandt, which is now more than 20 years old. The findings are very similar, and there is no mention or comparison here.

References:

- Hauert C, De Monte S, Hofbauer J, Sigmund K. Volunteering as red queen mechanism for cooperation in public goods games. *Science*. 2002 May 10;296(5570):1129-32.
- Brandt, Hannelore, Christoph Hauert, and Karl Sigmund. "Punishing and abstaining for public goods." *Proceedings of the National Academy of Sciences of the United States of America* 103.2 (2006): 495.
- Hauert C, De Monte S, Hofbauer J, Sigmund K. Replicator dynamics for optional public good games. *Journal of Theoretical Biology*. 2002 Sep 21;218(2):187-94.
- Hauert C, Traulsen A, Brandt H, Nowak MA, Sigmund K. Via freedom to coercion: the emergence of costly punishment. *science*. 2007 Jun 29;316(5833):1905-7.

Response: We appreciate the reviewer for linking our study with the important literature on the impact of introducing an option to abstain from groups (i.e., the loner option) in public goods provisioning. We regret that we failed to cite this highly relevant literature in the original manuscript. We have now incorporated the pertinent literature on the optional public goods game, originating from the work of Hauert and Brandt, in the Introduction.

(Lines 21 to 25) "Yet, such models have largely sidelined the fact that many collaborations in the real world do not involve the entire public or take place within predetermined group boundaries (but see literature of optional public goods game²²⁻²⁵ for important exceptions)."

Then, we have discussed the positioning of the current study in relation to the studies by Hauert and Brandt (and the subsequent studies). Although there is certainly a similarity between our study and theirs if summarized as a general statement such as "the loner option alleviates social dilemmas in public goods provisioning," we would like to note that the underlying mechanisms and dynamics differ in important ways. First, the series of studies by Hauert et al. assumed a linear public goods game (with a constant slope for the supply function of the public good) and explored a mechanism driven by the fact that the marginal per capita return increases as group size becomes smaller. That is, in their setup, if enough players choose to abstain, the group becomes so small that the game ceases to be a social dilemma anymore (i.e., cooperation yields greater payoff to the individuals themselves than defection). Second, their approach is rooted in evolutionary game theory, where the proportion

of agents adopting each strategy increases or decreases according to their fitness over iterations. There, the changes in the frequencies of strategies occur at the population level (i.e., as changes in the composition of agents following fixed strategies, but not because of behavioral changes that occur within each agent).

Our study diverges from the studies by Hauert and Brandt in both respects. First, we focus on a threshold, rather than a linear, public goods game. This distinction is not trivial, because here, the number of individuals choosing the loner option does NOT alleviate the social dilemma. In our framework, formed groups continue to face the issue of free riding even though loners voluntarily decide to be excluded from the goods they create. Second, different from their evolutionary game model, our model assumes decision-making agents who best respond to their own subjective beliefs, and we analyzed the empirical data accordingly. That is, agents can change their behaviors flexibly according to changes in their subjective beliefs. To sum up, our model showed how voluntary participation can aid collaboration under social dilemmas by facilitating the group formation of more optimistic individuals (please see also R.1.7 and R.2.0), and the experimental results are clearly in line with these predictions. We believe that these critical distinctions highlight the complementarity of our investigation to the optional public goods literature that originated from Hauert and Brandt.

In light of these discussions, we have added a paragraph in the Discussion to appropriately refer to and compare our work with that of Hauert et al. and Brandt et al.

(Lines 759 to 788) *“Another line of research that addresses the potential merits of individual outside options is the optional public goods game, most prominently introduced by Hauert, Brandt, and colleagues^{22–25}. They argued that as more individuals exit from the group where a linear public goods game is played, the return to a remaining player’s own contribution (marginal per capita return) increases to the point where cooperation is more beneficial than defection—if enough players choose to leave, the game ceases to be a social dilemma any more. Through evolutionary models, they showed that the population is not dominated by defectors but usually oscillates with cooperators, defectors, and loners coexisting in the long run. Note, however, in our setting with a threshold public goods game, the number of individuals choosing the individual option does not alleviate the social dilemma; groups still face the issue of free riding even though loners are voluntarily excluded from the public goods. Moreover, unlike the evolutionary analysis that concerns populational dynamics over time among agents each following a predetermined fixed strategy^{22–25}, our model assumes that agents change behaviors flexibly in response to changes in beliefs. We have empirically verified that participants indeed updated their subjective beliefs in response to the absence or presence of an individual option, even in one-shot decision scenarios. We believe that these critical distinctions highlight the complementarity of our investigation to the previous studies.”*

R.2.2 Another issue with the paper is that the model is not general enough. It only considers a small number of possible thresholds, and restricts itself to numerical results in a very small range of parameter. Some generality may make this contribution stronger.

Response: Your point that our model focuses on a limited range of parameters, including group size, the threshold value, and payoffs for the loner option, is important. In response, we have expanded our model analysis in the supplementary materials, relaxing some of the previous assumptions about group size, threshold value, and the loner's payoffs. Please refer to the revised Supplementary Information 4 concerning the additional analyses.

Additionally, we would like to note that the specific payoff structure in this study was not employed arbitrarily but was designed deliberately for the ease of participants' understanding. For instance, the loner option's payoff of 20 points was set to fall in the middle of the payoff for a failing group (0 or 10 points) and a successful group (30 or 40 points). This design was crucial to ensure that participants could easily grasp the game's incentive structure, thereby enabling rigorous experimental control.

We would also like to reiterate that our primary objective in modeling for this paper was not necessarily to develop a comprehensive model for the threshold public goods game with the loner option. Instead, our aim was to generate a priori predictions about what can happen empirically with and without the loner option. These predictions are crucial as they serve as a benchmark for the empirical data, providing reference points and constraints to interpret the experimental results.

Nonetheless, we do acknowledge that exploring a more general model, which encompasses various group sizes, payoff structures, and threshold levels, is an important extension for future research. Thus, besides some extensions reported in the revised Supplementary Information 4, we have added sentences about the limitation of our model in the Discussion, emphasizing the importance of gaining generality of our findings in the future.

(Lines 852 to 863)

*) For any line numbers from the main text cited in this document, please refer to the corresponding lines in the two-column version of the revised main manuscript file with all tracked changes accepted.

“Last, ours is not a general model, in that it does not comprehensively explore key parameters, including the group size, potential benefits of successful collaboration, and loner benefits. We partially extended our model analysis by relaxing the specific assumptions about these parameters employed in the experiment (see Supplementary Information 4 and Supplementary Figs. 4 and 5). These extensions yielded mostly the same results as in our original analysis, yet admittedly they may not be general enough. Carefully extrapolating the findings for broader

parameter regions would be a fruitful future direction.”

R.2.3 The paper also doesn't consider dynamics explicitly, which is important in order to endogenise ϕ . Altogether, I find the contribution marginal and better suited to a specialised journal once the appropriate comparisons with the work of Hauert et al, and Brandt et al. are considered.

Response: Thank you for pointing out the issue of dynamic belief formation, which we think parallels Reviewer 1's Comment 5 (R.1.5). Indeed, we did initially attempt to model players' belief formation. As we have discussed in our response to R.1.5 in detail, the process must be recursive in such a way that forming optimistic beliefs at one point could lead to pessimistic beliefs at the next point owing to the anticipation of prevalent free riding by optimistic others. Modeling this recursive updating of beliefs, however, would require introducing additional auxiliary parameters, such as the depth of players' thoughts and the noise in introspection or in expectation of others' decisions. Complicating matters further, when endogenizing the population-level distribution of beliefs ($\phi(\gamma)$), we may further need to consider population-level distributions of these parameters, as they are known to vary among individuals (e.g., Nagel, 1995; Coricelli & Nagel, 2009). Therefore, we restrict ourselves to reporting only the convergent cases (Nash equilibrium; see lines 276 to 291 and Supplementary Information 1) and acknowledging the difficulty in making strong a priori predictions in this regard.

Having said that, we do acknowledge your point that endogenizing the belief formation is preferable in terms of modeling and have now included the following paragraph in the Discussion (see also our response to R.1.5).

(Lines 826 to 841) *“Second, our model does not explicitly predict how individuals form their subjective beliefs. We first assume that these beliefs remain intact regardless of whether the group participation is voluntary or mandatory (Fig. 1B & C), and then examine equilibrium prediction, where beliefs ultimately converge with the distribution of players' actions (Supplementary Information 1). We took this approach because we found it intractable to explicitly model belief formation given that it depends heavily on additional individual factors. These factors include players' depth of thought and their perceptions of others' decision-making noise—aspects that are known to vary widely among individuals and contexts^{59–61}. Nevertheless, we acknowledge the potential value and interest in fully endogenizing players' beliefs.”*

References:

- Balliet, D. & Lindström, B. Inferences about interdependence shape cooperation. *Trends Cogn.*

Sci. **27**, 583–595 (2023).

- Coricelli, G. & Nagel, R. Neural correlates of depth of strategic reasoning in medial prefrontal cortex. *Proc. Natl. Acad. Sci. U. S. A.* **106**, 9163–9168 (2009).
- Gross, J., Veistola, S., De Dreu, C. K. W. & Van Dijk, E. Self-reliance crowds out group cooperation and increases wealth inequality. *Nat. Commun.* **11**, 5161 (2020).
- Nagel, R. Unraveling in guessing games: An experimental study. *Am. Econ. Rev.* **85**, 1313–1326 (1995).

Reviewer 3

R.3.0 I thought this was a terrific paper. The theorizing is lucid, the methods are rigorous, and the results are compelling. Although the idea of opting in vs. out has been examined elsewhere, I really like the analysis of individual beliefs.

Response: We thank the reviewer for the favorable evaluation and supporting the publication of our manuscript. For any line numbers from the main text cited in this document, please refer to the corresponding lines in the two-column version of the revised main manuscript file with all tracked changes accepted.

R.3.1 That said, I have several questions about loner's externality. First, given that this is perhaps the most novel insight (as italicized in the frontend of the paper, line 40), I am slightly disappointed that it was not examined more directly in the main experiment. Although I appreciate the theoretical discussion on p. 11-13, I would have liked to see, for instance, how loners' externality changes people's beliefs about group cooperation (y). Short of adding a new study, perhaps you could move the theoretical discussion to the top of the Results section and use it to motivate your experimental design (as opposed to ending the paper with it).

Response: Thank you very much for acknowledging the importance of our discussion about the varying degrees of loners' externality, and also the relevant suggestions regarding an additional experiment as well as moving the discussions in the Introduction. To be brief, we have now reported the results of an additional experiment that directly manipulated the loners' externality while retaining the order of discussions as it was. In the following, let us explain the reasons.

The original question we asked was whether, and if so how, an individual outside option can support collaborative success. We built a simple model and conducted an experiment directly manipulating the presence or absence of the individual option. Results show that the frequency of cooperators within groups increases under voluntary participation compared to mandatory participation via the two interconnected cognitive pathways (as reported in the main text). Since these results are by no means obvious a priori, and the (pre-registered) main experiment focused exclusively on this question, we have decided to retain how we motivate the study in the Introduction.

Importantly, however, in our original setting, we implicitly assumed that loners have no externality (impact) on groups and thus focused on $\#C/(\#C+\#D)$ as the effective cooperation rate (please see our responses to Reviewer 1's Comments R.1.1 and R.1.3). This aggregation cannot be justified if loners have an externality on group outcome (as in the case of Gross et al. [new ref. 31]).

We thus extended the aggregation of effective cooperation rate to incorporate varying degrees of loners' externality as a parameter $\rho \in [0, 1]$ (Eq. 4 in the main text) and argued that specific values of ρ reflect the degree of flexibility of group boundaries. Using the choice data from the main experiment, we then showed theoretically that introducing the outside individual option aids collaboration if loners have no externality but hinders collaboration with high externality (Fig. 4).

However, this original theoretical prospect is limited in the following sense. We based our calculation of the effective cooperation rate (Eq. 4) and group success rate (Fig. 4) on participants' decision data under $\rho = 0$ (i.e., the voluntary conditions of the experiment), while assuming that the exact proportion of each choice (i.e., #C, #D, and #L, respectively) remains the same across different ρ 's. However, different levels of $\rho (> 0)$ may affect participants' decisions per se, possibly lowering the effective cooperation rate (Eq. 4). In the original manuscript, for just this reason, we noted that the blue line in Fig. 4 should be seen as an upper bound. However, reflecting on your thoughtful comment, we realize that this point should be examined empirically to see the possible sensitivity of the effective cooperation rate and the resultant group success rate against the larger ρ . This led us to conduct an additional experiment directly manipulating loners' externality using a within-subject design. Employing the same protocol as in the voluntary condition of the main experiment with the threshold at 4, we manipulated the loners' externality at three levels ($\rho = 0, 0.5, \text{ or } 1$).

Results of the new experiment have clearly supported the original theoretical prospect (Fig. 5). The group success rate was close to 1 (88.4%) at $\rho = 0$ but decreased close to 0 (1.1%) at $\rho = 1$. Furthermore, we observed that the drops in cooperation and group success were already large at $\rho = 0.5$ (Fig. 5B). This drop was attributed not just to the aggregation method (Eq. 4) but also to a significant increase in the number of loners from 28.6% at $\rho = 0$ to 65.9% at $\rho = 0.5$ (Fig. 5A). These patterns suggest a new insight: that players' perception of loners' small externality at the time of decision making is another important factor for the individual option to support collaborative success. Interestingly, these differences in action selection across ρ values were accompanied by differences in beliefs about effective cooperation rates. Participants were significantly more optimistic about others' cooperation at smaller ρ values (Fig. 5C gray histogram; a mixed-effect regression: $\beta = -3.55$, CI $[-4.25, -2.85]$).

Accordingly, we have included these arguments and results from the new experiment in the revised Results.

(Lines 642 to 688) *“Note that the individual choice data used to construct Fig. 4B were obtained from the zero-externality scenario ($\rho=0$) of the voluntary conditions in the main experiment. Given that variations in the loners' externality are likely to affect not only the aggregation method (Eq. 4) but also the participants' action selections themselves, restriction of the individual choice data just under $\rho=0$ may affect the prospect of the theoretical analysis.*

To explore to what extent the group success rate is negatively affected by larger ρ , we

conducted an additional experiment. Employing the same protocol as in the voluntary condition of the main experiment with the threshold at 4, here, we manipulated the loners' externality at three levels ($\rho=0, 0.5, \text{ or } 1$; see Methods for details). The results are shown in Fig. 5. The participants indeed changed their action selections (cooperate, defect, or leave), responding to the differences in ρ . As ρ increased, the sheer number of participants choosing to cooperate or to defect decreased, while those opting for leaving increased (Fig. 5A). Notably, the resultant effective cooperation rate (Eq. 4) and the group success rate (Eq. 5) were higher at lower ρ values (Fig. 5B); very high group success (88.4%) at $\rho = 0$, which we observed in the main experiment (95.6%; Fig. 2B: threshold = 4), was replicated. We observed that the drops in cooperation and group success were already large at $\rho = 0.5$. This drop is attributed not just to the aggregation method (Eq. 4) but also to a significant increase in the number of loners from 28.6% (at $\rho = 0$) to 65.9% (at $\rho = 0.5$; Fig. 5A). These patterns suggest that players' perception of loners' small externality at the time of decision making is an important requirement for an individual option to support collaborative success. Also, of note, these differences in action selections across ρ values were accompanied by differences in beliefs about effective cooperation rates (Eq. 4); participants were significantly more optimistic about others' cooperativeness at smaller ρ values (Fig. 5C gray histogram; a mixed-effects logistic regression: $z = -16.24$, $\beta = -3.55$, $p < .001$, 95% CI $[-3.98, -3.14]$). Further, the mapping pattern from beliefs to decisions was also replicated, confirming the monotonic increase in cooperation rate and decrease in leaving rate as a function of beliefs about others' cooperativeness (Fig. 5C orange and green lines)."

R.3.2 Second, I am curious about how to conceptualize loners' externality as a continuous variable. If global versus local public goods games represent the extreme ends of the continuum, how should we think about intermediate cases? When do we see $\rho=0.5$ such that implementing individual options do not make any difference?

Response: We appreciate the reviewer for raising this important point. We fully agree that introducing ρ (a parameter conceptualizing the externality of loners on group outcomes) as a continuous variable has its full merit only if one can interpret the meaning of intermediate values between $\rho = 0$ and 1.

As discussed in the original manuscript, full externality ($\rho = 1$) corresponds to where group boundaries are fixed from the outset irrespective of the individuals' decision to opt for a collective or individual solution. Zero externality ($\rho = 0$) corresponds to where groups are flexibly formed by non-loners after loners' decisions. The intermediate cases ($0 < \rho < 1$) between these two extremes correspond to where loners partially get involved in groups. For example, when a group member

expresses an intention to leave, he or she may not be immediately replaced by a new entrant and may end up remaining in the group temporarily, possibly because of the group's limited ability to recruit others or even as a part of a formal contract. In the additional experiment, we implemented $\rho = 0.5$ as an intermediate case (see lines 980 to 1030 in Methods). As stated in R.3.1, we observed that the drops in cooperation and group success were already large at $\rho = 0.5$ (Fig. 5B), resulting from a significant increase in the number of loners (from 28.6% at $\rho = 0$ to 65.9% at $\rho = 0.5$; Fig. 5A) and a significant decrease in optimism about others' cooperation at larger ρ values (Fig. 5C).

We have amended a paragraph in the Results about the interpretation and real-life examples of the intermediate level of loners' externality.

(Lines 620 to 627) *“Similarly, there should be cases where loners exert a partial externality on the group outcome. For example, when group members express an intention to leave, they may not be immediately replaced by new entrants and may end up remaining in the group temporarily, possibly because of the group's limited ability to recruit others or even as a part of a formal contract. These cases correspond to intermediate values of ρ ($0 < \rho < 1$).”*

R.3.3 Third, I am a bit confused about the terms global and local. In the global case, public goods games are still played at the “local” level of predefined groups. Would it make sense to just call these cases opting in vs. out?

Response: Thank you for pointing out the lack of clarity in explaining the determinants of the varying degrees of loners' externality on the collective outcome. Reflecting on your comments, we have realized that the global/local dichotomy is not appropriate. We now believe that the determinant of loners' externality is better summarized as the flexibility of the group boundaries in question: If group boundaries are fixed from the outset irrespective of the individuals' decision to opt for a collective or individual solution, loners have full externality on the group outcome (the global case, such as climate change, can be an instance of fixed boundaries, since by definition everyone must be involved). In contrast, if groups are formed flexibly by non-loners after loners' decisions, the externality is minimal. Although we think opting-in vs. opting-out reflects the same distinction by implicitly assuming different timing of group formation, we prefer the term fixed and flexible for the sake of clarity (especially for non-native English speakers).

Thus, in the revised manuscript, we have removed the dichotomy of global versus local public goods and instead explicitly discuss the flexibility of group boundaries. We have also revised Fig. 4A accordingly.

(Lines 544 to 547) *“Here, we propose an integrative view highlighting the varying degrees of externality (impact) that loners, who opt for individual solutions, still have on the outcome of group endeavors.”*

(Lines 571 to 597) *“Notice that, in their set-up, with more loners who choose the individual solution, a smaller number of villagers must share the cost to build the public dam that surrounds the entire village including the loners’ houses; consequently, in terms of provision of the public dam, loners function essentially the same as defectors who retain their endowment and do not fund any dam. That is, since group boundaries are fixed from the outset irrespective of the individuals’ decision to opt for a collective or individual solution, the cooperation rate determining the collective outcome is the proportion of cooperators among the entire population in the village (fixed group).*

In contrast, in our scenario focusing on collaboration², the cooperation rate determining the collective outcome for groups is the proportion of cooperators among players who opt in to groups. This is the natural consequence of our assumption that participation in collaboration is voluntary rather than mandatory. Group boundaries are not fixed from the outset but flexible, and group formation comes after the individuals’ decision to opt in or not; loners are excluded from the groups at the time of their formation. Note that this does not imply that people no longer suffer from the free-riding problem: Groups still must create shared benefits from the costly efforts of some members.”

(Fig. 4 legend) *“(A) Left: When groups are fixed in advance (e.g., the entire population is a group), loners have full externality on the collective outcome and effectively function the same as defectors to the group (e.g., the “village” example in ref. 31). Right: When group boundaries are flexible, groups consist only of individuals who voluntarily opt in, and thus loners have no externality on the collective outcome.”*

R.3.4 Line 242: This is a nice, punchy line, but it should say “Voluntary participation does just that.”

Response: Thank you, we have corrected the sentence and moved it to the end of the main text.

(Lines 872 to 873) *“Voluntary participation does just that.”*

Reviewers' Comments:

Reviewer #1:

Remarks to the Author:

The authors, in my view, did a great job in addressing all the comments and suggestions. The revisions and additional results, I think, make the paper much stronger and provide a compelling framework to think about opting-in / opting-out dynamics in cooperation/coordination problems.

There is only one thing I personally disagree with, pertaining to a sentence in the discussion: "We argue that in most mundane instances under voluntary participation, group collaboration neither involves the entire population nor occurs within predetermined group boundaries; groups form through voluntary participation and consist exclusively of individuals who voluntarily opt in. Consequently, loners have little to no externality on group endeavors."

While I tend to agree that one can think of many examples where groups form dynamically after the decision to opt-in / opt-out, especially on a societal level, this may not be the case and that is why many societies seem to restrict or heavily codify opting-out possibilities (e.g., private healthcare, opting out of paying taxes, private schooling etc.).

From my view, the strength of the model is that it could be taken as a framework to understand (on an abstract level) under which conditions groups may fare better to allow for opting-out and under which conditions groups may have an interest to restrict it (especially when also considering between-agent inequality, which is not considered).

Maybe that can be discussed a bit more nuanced. But this is also just an opinion and not part of the empirical test of the paper or the general scope.

As such, I just leave that as a suggestion and have no further comments. In my assessment this is a great paper that, although the model could be more general and the basic idea is not entirely new, the combination of a very clear model, interesting empirical evidence that even goes beyond the model (i.e., the two-path results), and some interesting parameters that can integrate disparate findings on optional play makes this a very valuable contribution to the literature.

Reviewer #2:

Remarks to the Author:

I want to thank the authors for taking the time to reply to all of my concerns in detail. I believe the changes in the manuscript are a step in the right direction and actually do a better job of explaining the nuances of the approach, its predictive power as well as its limitations.

My main takeaway is that this paper should actually be presented primarily as an empirical paper that does use a simple model and not so much as a modelling exercise with some empirical work adjacent.

The response reads:

"Although there is certainly a similarity between our study and theirs if summarized as a general statement such as *the loner option alleviates social dilemmas in public goods provisioning,* we would like to note that the underlying mechanisms and dynamics differ in important ways"

I think the nuance about mechanisms and dynamics needs to be reflected in the positioning of the paper and the title. The title certainly reads as the loner option alleviating social dilemmas. I think the nuances in the results are important. This work appears to me to be incremental in that sense, and therefore I still question whether the results are general enough to be of interest to the general public, being perhaps better suited to a more specialised journal.

The paper does offer some very interesting results. I liked and enjoyed the combination of experimental work and modelling. Once the nuances have been properly discussed (as they are in the revision) and the literature includes previous work on optional PGG models, the paper is sound, yet somehow incremental from my perspective as a modeller. Judging whether this is of general interest enough is probably an editorial decision at this stage.

Reviewer #3:

Remarks to the Author:

Thank you for attending to the reviewer feedback so carefully. I am impressed by the responses, and pleased that the authors went above and beyond to run the additional study that I suggested but did not think was critical.

Well done for an exemplary revision.

Reviewer 1

R.1.0

The authors, in my view, did a great job in addressing all the comments and suggestions. The revisions and additional results, I think, make the paper much stronger and provide a compelling framework to think about opting-in / opting-out dynamics in cooperation / coordination problems.

Response: Thank you very much. We greatly appreciate the thorough comments from the reviewers throughout this process.

R.1.1 There is only one thing I personally disagree with, pertaining to a sentence in the discussion: "We argue that in most mundane instances under voluntary participation, group collaboration neither involves the entire population nor occurs within predetermined group boundaries; groups form through voluntary participation and consist exclusively of individuals who voluntarily opt in. Consequently, loners have little to no externality on group endeavors."

While I tend to agree that one can think of many examples where groups form dynamically after the decision to opt-in / opt-out, especially on a societal level, this may not be the case and that is why many societies seem to restrict or heavily codify opting-out possibilities (e.g., private healthcare, opting out of paying taxes, private schooling etc.).

From my view, the strength of the model is that it could be taken as a framework to understand (on an abstract level) under which conditions groups may fare better to allow for opting-out and under which conditions groups may have an interest to restrict it (especially when also considering between-agent inequality, which is not considered). Maybe that can be discussed a bit more nuanced. But this is also just an opinion and not part of the empirical test of the paper or the general scope.

As such, I just leave that as a suggestion and have no further comments. In my assessment this is a great paper that, although the model could be more general and the basic idea is not entirely new, the combination of a very clear model, interesting empirical evidence that even goes beyond the model (i.e., the two-path results), and some interesting parameters that can integrate disparate findings on optional play makes this a very valuable contribution to the literature.

Response: Thank you very much for noting this important point. As you nicely summarized, we also believe that the main strength of our model, particularly in the latter part of the paper, is its ability to analyze the determinants of when individual options aid or hinder collaborative efforts. While we still think that many new collaborative opportunities that we encounter in our daily lives, particularly in their initiation phase, are flexible enough to minimize the externality of loners, we agree with you that some of the most significant societal challenges, such as combating climate change and sustaining the healthcare system, inherently have inflexible boundaries. We have incorporated these discussions into the Discussion section as follows.

(Lines 388 to 392) *“Certainly, some of the large-scale societal challenges, such as combating climate change and sustaining the healthcare system in a country, inevitably involve everyone in the society and thus have inflexible boundaries. However, we note that many new collaborative opportunities that we encounter in our daily lives, neither involve the entire population nor occur within predetermined group boundaries; groups form through voluntary participation and consist exclusively of individuals who voluntarily opt in. Consequently, loners have little to no externality on such group endeavors.”*

Reviewer 2

R.2.0 I want to thank the authors for taking the time to reply to all of my concerns in detail. I believe the changes in the manuscript are a step in the right direction and actually do a better job of explaining the nuances of the approach, its predictive power as well as its limitations.

Response: Thank you very much for reading our revised manuscript carefully and appreciating our revisions.

R.2.1 My main takeaway is that this paper should actually be presented primarily as an empirical paper that does use a simple model and not so much as a modelling exercise with some empirical work adjacent.

The response reads:

"Although there is certainly a similarity between our study and theirs if summarized as a general statement such as *the loner option alleviates social dilemmas in public goods provisioning,* we would like to note that the underlying mechanisms and dynamics differ in important ways"

I think the nuance about mechanisms and dynamics needs to be reflected in the positioning of the paper and the title. The title certainly reads as the loner option alleviating social dilemmas. I think the nuances in the results are important. This work appears to me to be incremental in that sense, and therefore I still question whether the results are general enough to be of interest to the general public, being perhaps better suited to a more specialised journal.

Response: Thank you for giving us the opportunity to reconsider the nuanced positioning of our paper in the literature, especially in relation to modeling work with similar conclusions (that social dilemmas are easier to solve with the loner option than without).

First, we acknowledge that our model is not necessarily general enough to state that we *theoretically show* how individual options facilitate collaboration. Instead, as has been better clarified in the revision process, the model mainly serves as a benchmark to predict and interpret experimental results. We have revised the pertinent sentences in the Introduction, Results, and Discussion sections accordingly.

Second, in our view, an even more relevant literature than the modeling work is that by Gross and colleagues. This is because their (and our) focus is on investigating how collaborative efforts in our

society are affected by the existence of individualistic alternatives. Thus, we and they share a primarily focus on people's actual responses rather than predictions from normative models, although the conclusions are somewhat divergent. To clarify these key contributions of our work, we have modified the title: "An outside individual option increases optimism and facilitates collaboration when groups form flexibly".

R.2.2 The paper does offer some very interesting results. I liked and enjoyed the combination of experimental work and modelling. Once the nuances have been properly discussed (as they are in the revision) and the literature includes previous work on optional PGG models, the paper is sound, yet somehow incremental from my perspective as a modeller. Judging whether this is of general interest enough is probably an editorial decision at this stage.

Response: Thanks to the thoughtful comments of the reviewer(s), we believe that our paper has been greatly improved in its fitness for the general audience of the journal.

Reviewer 3

R.3.0 Thank you for attending to the reviewer feedback so carefully. I am impressed by the responses, and pleased that the authors went above and beyond to run the additional study that I suggested but did not think was critical.

Well done for an exemplary revision.

Response: We thank the reviewer for the kind words. We believe that the paper has been greatly improved thanks to your valuable suggestion about the additional experiment.